# Timing of TORC1 inhibition dictates Pol III involvement in *Caenorhabditis elegans* longevity

Yasir Malik[1], Isabel Goncalves Silva[1], Rene Rivera Diazgranados[1], Colin Selman[3], Nazif Alic[2], Jennifer MA Tullet[1]

Organismal growth and lifespan are inextricably linked. Target of Rapamycin (TOR) signalling regulates protein production for growth and development, but if reduced, extends lifespan across species. Reduction in the enzyme RNA polymerase III, which transcribes tRNAs and 5S rRNA, also extends longevity. Here, we identify a temporal genetic relationship between TOR and Pol III in *Caenorhabditis elegans*, showing that they collaborate to regulate progeny production and lifespan. Interestingly, the lifespan interaction between Pol III and TOR is only revealed when TOR signaling is reduced, specifically in adulthood, demonstrating the importance of timing to control TOR regulated developmental versus adult programs. In addition, we show that Pol III acts in *C. elegans* muscle to promote both longevity and healthspan and that reducing Pol III even in late adulthood is sufficient to extend lifespan. This demonstrates the importance of Pol III for lifespan and age-related health in adult *C. elegans*.

## Introduction

In eukaryotic cells, three multi-subunit RNA polymerase (Pol) enzymes, Pol I, II, III, transcribe the nuclear genome. Each polymerase synthesises a distinct set of genes: Pol II transcribes all coding genes to generate mRNA, whereas polymerases Pol I and III only transcribe non-coding genes (1). RNA Pol III comprises 17 subunits and is specialised for the transcription of short, abundant, non-coding RNA transcripts. Its main gene products are tRNAs and 5S ribosomal RNA (rRNA), which are abundantly transcribed and required for protein synthesis (2). Other Pol III transcripts include small nuclear (snRNAs) and small nucleolar (snoRNAs), both of which are implicated in RNA processing, the 7SK snRNAs that plays role in regulating transcription, as well as other small RNAs including Y RNA, L RNA, vault RNA, and U6 spliceosomal RNA (3, 4) Pol III-mediated transcription regulates a wide range of biological processes including cell and organismal growth (5, 6, 7, 8), the cell cycle (9), differentiation (8, 10, 11), development (12), regeneration (13) and cellular responses to stress (14). Pol III subunits have also been implicated in a wide variety of disease states and lifespan (15, 16, 17).

Reducing the activity of Pol III by partially down-regulating its individual subunits can significantly extend lifespan in multiple model organisms including baker's yeast *Saccharomyces cerevisiae*, the nematode *Caenorhabditis elegans* and the fruit fly *Drosophila melanogaster* (18). In *C. elegans*, *rpc-1* encodes the largest Pol III subunit and its down-regulation by RNAi increases lifespan (18). In *D. melanogaster*, lifespan is increased in heterozygous females with a single functional copy of gene encoding for Pol III-specific subunit C53. Pol III activity in the fly gut limits survival and can drive ageing, specifically from the gut stem-cell compartment. Importantly, in flies, Pol III knockdown is downstream of the target of Rapamycin (TOR) pathway, an evolutionarily conserved regulator of longevity (18, 19)

The TOR kinase is a serine/threonine kinase that regulates growth and development chiefly by modulating protein synthesis in response to changes in nutrients and other cues (19) (Fig 1A). TOR can be part of two complexes, and it is specifically the TOR complex 1 (TORC1) that drives growth, development, and anabolic metabolism (19, 20). In *C. elegans*, complete loss of *let-363*, the gene which encodes the TOR ortholog (CeTOR), is lethal, whereas depletion of *let-363*/*CeTOR* by RNAi or pharmacologically extends lifespan (21) (Fig 1A).

Here we show that CeTOR and Pol III act together to control development and adult lifespan. Reducing CeTOR signaling from embryonic stages together with Pol III inhibition demonstrated their genetic interaction to control progeny production, but their ability to collaborate and regulate lifespan is only revealed when CeTOR signaling is reduced specifically in adulthood. This demonstrates the critical temporal importance of TOR signaling throughout life, how this alters its interactions with key transcriptional regulators, and the impact of this on developmental versus longevity programs. In addition, we show that Pol III acts in *C. elegans* muscle to promote longevity, healthspan and body-wall mitochondrial networks, and that late-life Pol III knockdown is sufficient to extend lifespan. Overall, our data uncover a mechanism which describes the interactions of TORC1 and Pol III in

[1]Division of Natural Sciences, School of Biosciences, University of Kent, Canterbury, Kent   [2]UCL Department of Genetics, Evolution & Environment, Institute of Healthy Ageing, London, UK   [3]Institute of Biodiversity, Animal Health and Comparative Medicine, University of Glasgow, Glasgow, Scotland

Correspondence: j.m.a.tullet@kent.ac.uk

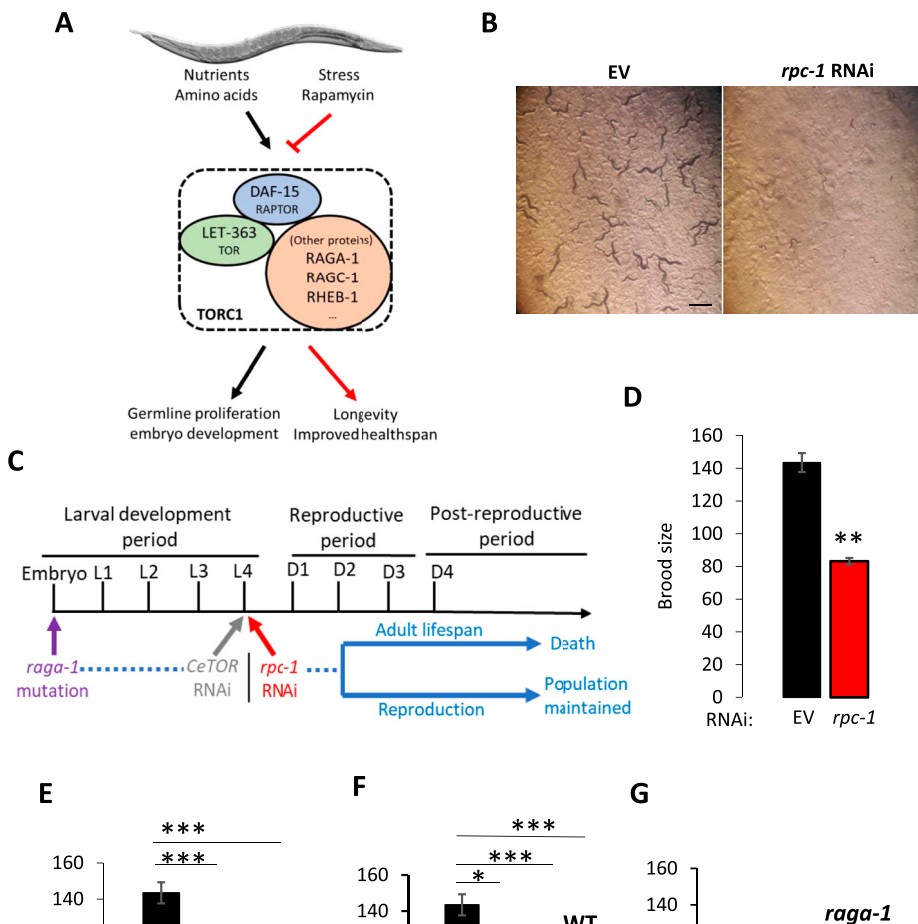

**Figure 1. CeTOR signalling and RNA Polymerase III interact to control reproduction.**
**(A)** Schematic showing the *C. elegans* CeTORC1 complex and its potential downstream outcomes in response to nutrient cues. **(B)** *rpc-1* RNAi leads to L3 stage developmental arrest. This RNAi treatment typically leads to a 60% reduction in *rpc-1* mRNA levels ([18]). Scale bar indicates 1 mm. **(C)** Schematic showing timings of the *rpc-1* and CeTOR interventions: for *rpc-1* and *let-363/CeTOR* RNAi worms were raised on OP50 and moved to the appropriate RNAi at the L4 stage; loss-of-function mutation in *raga-1(ok386)* is present throughout life, from the embryonic stage. **(D, E, F, G)** Interaction between the CeTOR pathway and *rpc-1* for brood size. Combined data for three biological replicates are shown. EV, Empty Vector. Statistical comparisons were made by One-way ANOVA. *P*-value: *$P < 0.05$, **$P < 0.005$, ***$P < 0.0005$; ns, non-significant.

reproduction and longevity, and the importance of Pol III for lifespan and age-related health in adult *C. elegans*.

# Results

## RNA polymerase III is important for *C. elegans* reproduction

RNA polymerase III is essential for growth and development ([6], [7], [22]). Indeed, when we reduced Pol III mRNA expression by feeding *C. elegans* bacteria expressing a dsRNA for *rpc-1* from the embryonic stage, the worms exhibited growth defects and more than 90% of worms were arrested at L2-L3 larval stage, whereas all animals in control RNAi developed to adults within 3 d (Fig 1B). This

confirms the requirement of Pol III for reproduction has not been examined. To determine the effect of Pol III knockdown on progeny production, we raised the worms to early adulthood on the standard laboratory food source *Escherichia coli* OP50-1, transferring to *rpc-1* RNAi at the L4 larval stage (Fig 1C). We found that *C. elegans* treated with *rpc-1* RNAi at this late larval stage developed into adults but displayed a dramatic reduction in their total brood size compared to control (~40% $P = 2.98 \times 10^{-6}$, Fig 1D). The reduction in brood size is likely independent of spermatogenesis as *rpc-1* RNAi did not diminish the total number of sperm in self-fertilized hermaphrodites (Fig S1A and B). Nor did *rpc-1* RNAi treatment lead to any change in embryonic viability, or age-specific fecundity and embryos hatched into larvae within a similar time frame as WT (Figs S1C and S2A). This suggests that Pol III inhibition from late larval

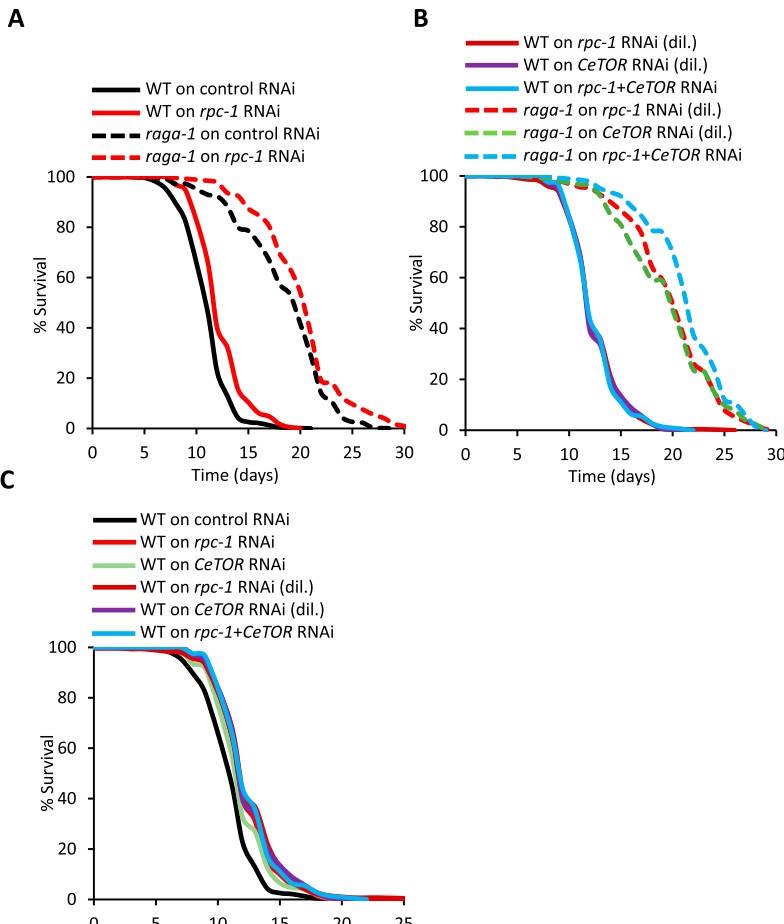

**Figure 2. CeTOR signalling and RNA Polymerase III interact temporally to control lifespan.**
**(A)** *rpc-1* RNAi and *raga-1(ok386)* additively increases lifespan. **(B)** *let-363/CeTOR* RNAi and *raga-1(ok386)* mutation extend lifespan and are additive with *rpc-1* RNAi. **(C)** *let-363/CeTOR* RNAi interacts with *rpc-1* RNAi to extend lifespan. **(A, B, C)** Diluted and non-diluted *let-363/CeTOR* or *rpc-1* RNAi treatments extend lifespan equally (Table 1). One representative experiment is shown, refer to Table 1 for data on all replicates and statistical analysis.

development is sufficient to reduce progeny production without affecting spermatogenesis or embryo maturation *ex-utero*. Taken together, these data show that Pol III is required for normal organismal growth and reproductive processes.

## TOR signaling interacts genetically with RNA polymerase III to control *C. elegans* reproduction

Reproduction and embryonic development in worms is an intense anabolic process requiring high levels of protein translation (23, 24). The TORC1 pathway acts as a master switch for governing cellular growth, whereas Pol III transcribes two of the most important noncoding RNA families (tRNAs and rRNAs) required for protein synthesis (2). To test whether Pol III could act downstream of TORC1 to control reproduction, we examined the epistatic relationship between *rpc-1* knockdown and *C. elegans* TORC1 (CeTORC1) using progeny production as a readout. In *C. elegans*, the gene encoding RagA (a RAS-related GTP-binding protein) is *raga-1/RagA*, which acts as a positive regulator of the CeTORC1 kinase (25) (Fig 1A). Animals carrying the *raga-1(ok386)* mutation have constitutively low CeTORC1 signaling throughout development and adulthood (26) (Fig 1C). *raga-1(ok386)* mutants develop normally, but adults have a reduced brood size compared with WT (Fig 1E, $P = 4.81 \times 10^{-6}$).

However, treatment of *raga-1(ok386)* mutants with *rpc-1* RNAi resulted in a further reduction in progeny numbers compared with either *raga-1/RagA* mutation or *rpc-1* RNAi alone (Fig 1E, $P = 7.91 \times 10^{-11}$ and $P = 6.67 \times 10^{-8}$, respectively).

The *raga-1(ok386)* strain harbors a mutation predicted to remove the entire gene, and is considered null. However, TORC1 is controlled by multiple inputs, and we were concerned that the *raga-1(ok386)* mutation was not sufficient to completely reduce CeTORC1 signaling. To address this, we depleted the *C. elegans* TOR kinase itself using RNAi. *C. elegans* fed *let-363/CeTOR* RNAi from the L4 stage had a reduced brood size compared with control (Fig 1F, $P = 4.9 \times 10^{-4}$). However, combining *let-363/CeTOR* RNAi with *rpc-1* RNAi was non-additive compared with either *rpc-1* or *let-363/CeTOR* RNAi (Fig 1F, $P = 0.57$ and $P = 0.62$, respectively). This was not because of an artifact of performing double RNAi as diluting either *let-363/CeTOR* or *rpc-1* RNAi by 50% with EV control achieved the same fecundity-related phenotypes as undiluted RNAi (Fig S2A–C). Thus, this implies that Pol III and TOR act in the same pathway to regulate *C. elegans* reproduction. To explore this epistatic relationship further, we examined progeny production in *raga-1(ok386)* mutants treated with *let-363/CeTOR* RNAi. Depletion of *let-363/CeTOR* from early adulthood resulted in a further reduction in *raga-1(ok386)* brood size (Fig 1G, $P = 1.31 \times 10^{-5}$), indicative that the initial *raga-1/RagA*

**Table 1.  Lifespan data for Fig 2.**

| Figure | Trial | Strain | Genotype | Mean lifespan (days) | Extension (%) raga-1 control | Extension (%) WT control | P-value (Log-rank) vs | N dead (total) |
|---|---|---|---|---|---|---|---|---|
| 2C | 1 | WT | Control RNAi | 10.83 | | | | 93 |
| 2C | | WT | rpc-1 RNAi | 11.72 | | 8.22 | WT<0.001 | 91 |
| 2C | | WT | let-363/CeTOR RNAi | 11.52 | | 6.37 | WT<0.05<br>WT rpc-1:NS | 95 |
| 2B, 2C | | WT | rpc-1 RNAi diluted (1:1 with control) | 11.73 | | 8.31 | WT<0.001<br>WT rpc-1:NS<br>WT let-363/ceTOR:NS | 91 |
| 2B, 2C | | WT | let-363/CeTOR RNAi diluted (1:1 with control) | 11.84 | | 9.33 | WT<0.001<br>WT rpc-1:NS<br>WT let-363/ceTOR:NS<br>WT rpc-1:NS | 72 |
| 2B, 2C | | WT | rpc-1 RNAi + let-363/CeTOR RNAi | 12.4 | | 14.50 | WT<0.001<br>WT rpc-1<0.05<br>WT let-363/ceTOR<0.05<br>WT rpc-1 :NS<br>WT let-363/ceTOR :NS | 84 |
| 2A | | raga-1(ok386) | Control RNAi | 20.38 | | 70.83 | WT<0.001<br>WT rpc-1<0.001<br>WT let-363/ceTOR<0.001<br>WT rpc-1 <0.001<br>WT let-363/ceTOR <0.001<br>WT rpc-1 + let-363/CeTOR<0.001 | 90 |
| 2A | | raga-1(ok386) | rpc-1 RNAi | 21.9 | 7.46 | 83.57 | WT<0.001<br>WT rpc-1<0.001<br>WT let-363/ceTOR<0.001<br>WT rpc-1 diluted<0.001<br>WT let-363/ceTOR diluted <0.001<br>WT rpc-1 + let-363/CeTOR<0.001<br>raga-1<0.05 | 81 |
| 2B | | raga-1(ok386) | let-363/CeTOR RNAi | 22.03 | 8.10 | 84.66 | WT<0.001<br>WT rpc-1<0.001<br>WT let-363/ceTOR<0.001<br>WT rpc-1 diluted<0.001<br>WT let-363/ceTOR diluted<0.001<br>WT rpc-1 + let-363/CeTOR<0.001<br>raga-1<0.05<br>raga-1 rpc-1:NS | 80 |
| 2B | | raga-1(ok386) | rpc-1 RNAi diluted (1:1 with control) | 21.84 | 7.16 | 83.07 | WT<0.0001<br>WT rpc-1<0.0001<br>WT let-363/ceTOR<0.0001<br>WT rpc-1 diluted<0.0001<br>WT let-363/ceTOR diluted <0.0001<br>WT rpc-1 + let-363/CeTOR<0.0001<br>raga-1<0.05<br>raga-1 rpc-1:NS<br>raga-1 ceTOR:NS | 81 |

**Table 1. Continued**

| Figure | Trial | Strain | Genotype | Mean lifespan (days) | Extension (%) raga-1 control | Extension (%) WT control | P-value (Log-rank) vs | N dead (total) |
|---|---|---|---|---|---|---|---|---|
| 2B | | *raga-1(ok386)* | *rpc-1* RNAi + *let-363/CeTOR* RNAi | 23.14 | 13.60 | 113.94 | WT<0.0001<br>WT *rpc-1*<0.0001<br>WT *let-363/ceTOR*<0.0001<br>WT *rpc-1* diluted<0.0001<br>WT *let-363/ceTOR* diluted <0.0001<br>WT *rpc-1* + *let-363/CeTOR*<0.0001<br>*raga-1* control<0.05<br>*raga-1* rpc-1 dil.<0.05<br>*raga-1* ceTOR dil.<0.05<br>WT *rpc-1*<0.0001 | 82 |
| | 2 | WT | Control RNAi | 10.55 | | | | 88 |
| | | WT | *rpc-1* RNAi | 11.77 | | 10.37 | WT<0.001 | 93 |
| | | WT | *let-363/CeTOR* RNAi | 11.89 | | 11.27 | WT<0.05<br>WT *rpc-1*:NS | 100 |
| | | WT | *rpc-1* RNAi diluted (1:1 with control) | 11.51 | | 8.34 | WT<0.001<br>WT *rpc-1*:NS<br>WT *let-363/ceTOR*:NS | 89 |
| | | WT | *let-363/CeTOR* RNAi diluted (1:1 with control) | 11.40 | | 7.46 | WT<0.001<br>WT *rpc-1*:NS<br>WT *let-363/ceTOR*:NS<br>WT *rpc-1*:NS | 81 |
| | | WT | *rpc-1* RNAi + *let-363/CeTOR* RNAi | 12.43 | | 15.12 | WT<0.001<br>WT *rpc-1*<0.05<br>WT *let-363/ceTOR*<0.05<br>WT *rpc-1* :NS<br>WT *let-363/ceTOR* :NS | 94 |
| | | *raga-1(ok386)* | Control RNAi | 19.98 | | 89.38 | WT<0.001<br>WT *rpc-1*<0.001<br>WT *let-363/ceTOR*<0.001<br>WT *rpc-1* <0.001<br>WT *let-363/ceTOR* <0.001<br>WT *rpc-1* + *let-363/CeTOR*<0.001 | 78 |
| | | *raga-1(ok386)* | *rpc-1* RNAi | 22.01 | 10.11 | 108.62 | WT<0.001<br>WT *rpc-1*<0.001<br>WT *let-363/ceTOR*<0.001<br>WT *rpc-1* diluted<0.001<br>WT *let-363/ceTOR* diluted <0.001<br>WT *rpc-1* + *let-363/CeTOR*<0.001<br>*raga-1*<0.05 | 91 |
| | | *raga-1(ok386)* | *rpc-1* RNAi diluted (1:1 with control) | 21.88 | 9.52 | 108.34 | WT<0.001<br>WT *rpc-1*<0.001<br>WT *let-363/ceTOR*<0.001<br>WT *rpc-1* diluted<0.001<br>WT *let-363/ceTOR* diluted<0.001<br>WT *rpc-1* + *let-363/CeTOR*<0.001<br>*raga-1*<0.05<br>*raga-1 rpc-1*:NS | 94 |

| Figure | Trial | Strain | Genotype | Mean lifespan (days) | Extension (%) *raga-1* control | Extension (%) WT control | *P*-value (Log-rank) vs | *N* dead (total) |
|---|---|---|---|---|---|---|---|---|
| | | *raga-1(ok386)* | CeTOR RNAi diluted (1:1 with control) | 22.21 | 11.16 | 110.52 | WT<0.0001<br>WT *rpc-1*<0.0001<br>WT *let-363/ceTOR*<0.0001<br>WT *rpc-1* diluted<0.0001<br>WT *let-363/ceTOR* diluted <0.0001<br>WT *rpc-1* + *let-363/CeTOR*<0.0001<br>*raga-1*<0.05<br>*raga-1* rpc-1:NS<br>*raga-1* ceTOR:NS | 96 |
| | | *raga-1(ok386)* | rpc-1 RNAi + let-363/CeTOR RNAi | 23.08 | 15.51 | 118. | WT<0.0001<br>WT *rpc-1*<0.0001<br>WT *let-363/ceTOR*<0.0001<br>WT *rpc-1* diluted<0.0001<br>WT *let-363/ceTOR* diluted <0.0001<br>WT *rpc-1* + *let-363/CeTOR*<0.0001<br>*raga-1* control<0.05<br>*raga-1* rpc-1 dil.<0.05<br>*raga-1* ceTOR dil.<0.05 | 91 |
| | 3 | WT | Control RNAi | 10.69 | | | | 67 |
| | | WT | rpc-1 RNAi | 11.72 | | 11.09 | WT<0.001 | 74 |
| | | WT | let-363/CeTOR RNAi | 11.88 | | 12.60 | WT<0.05<br>WT *rpc-1*:NS | 78 |
| | | WT | rpc-1 RNAi diluted (1:1 with control) | 11.67 | | 10.61 | WT<0.001<br>WT *rpc-1*:NS<br>WT *let-363/ceTOR*:NS | 81 |
| | | WT | let-363/CeTOR RNAi diluted (1:1 with control) | 11.63 | | 10.23 | WT<0.001<br>WT *rpc-1*:NS<br>WT *let-363/ceTOR*:NS<br>WT *rpc-1*:NS | 83 |
| | | WT | rpc-1 RNAi + let-363/CeTOR RNAi | 11.97 | | 13.45 | WT<0.001<br>WT *rpc-1*<0.05<br>WT *let-363/ceTOR*<0.05<br>WT *rpc-1* :NS<br>WT *let-363/ceTOR* :NS | 76 |
| | | *raga-1(ok386)* | Control RNAi | 19.11 | | 81.13 | WT<0.001<br>WT *rpc-1*<0.001<br>WT *let-363/ceTOR*<0.001<br>WT *rpc-1* <0.001<br>WT *let-363/ceTOR* <0.001<br>WT *rpc-1* + *let-363/CeTOR*<0.001 | 69 |
| | | *raga-1(ok386)* | rpc-1 RNAi | 22.15 | 15.90 | 109.95 | WT<0.001<br>WT *rpc-1*<0.001<br>WT *let-363/ceTOR*<0.001<br>WT *rpc-1* diluted<0.001<br>WT *let-363/ceTOR* diluted <0.001<br>WT *rpc-1* + *let-363/CeTOR*<0.001<br>*raga-1*<0.05 | 71 |

**Table 1. Continued**

| Figure | Trial | Strain | Genotype | Mean lifespan (days) | Extension (%) raga-1 control | Extension (%) WT control | P-value (Log-rank) vs | N dead (total) |
|---|---|---|---|---|---|---|---|---|
| | | raga-1(ok386) | rpc-1 RNAi diluted (1:1 with control) | 21.56 | 12.82 | 104.36 | WT<0.001 | 67 |
| | | | | | | | WT rpc-1<0.001 | |
| | | | | | | | WT let-363/ceTOR<0.001 | |
| | | | | | | | WT rpc-1 diluted<0.001 | |
| | | | | | | | WT let-363/ceTOR diluted<0.001 | |
| | | | | | | | WT rpc-1 + let-363/CeTOR<0.001 | |
| | | | | | | | raga-1<0.05 | |
| | | | | | | | raga-1 rpc-1:NS | |
| | | raga-1(ok386) | CeTOR RNAi diluted (1:1 with control) | 21.38 | 11.87 | 102.65 | WT<0.0001 | 77 |
| | | | | | | | WT rpc-1<0.0001 | |
| | | | | | | | WT let-363/ceTOR<0.0001 | |
| | | | | | | | WT rpc-1 diluted<0.0001 | |
| | | | | | | | WT let-363/ceTOR diluted <0.0001 | |
| | | | | | | | WT rpc-1 + let-363/CeTOR<0.0001 | |
| | | | | | | | raga-1<0.05 | |
| | | | | | | | raga-1 rpc-1:NS | |
| | | | | | | | raga-1 ceTOR:NS | |
| | | raga-1(ok386) | rpc-1 RNAi + let-363/CeTOR RNAi | 23.78 | 24.43 | 125.40 | WT<0.0001 | 78 |
| | | | | | | | WT rpc-1<0.0001 | |
| | | | | | | | WT let-363/ceTOR<0.0001 | |
| | | | | | | | WT rpc-1 diluted<0.0001 | |
| | | | | | | | WT let-363/ceTOR diluted <0.0001 | |
| | | | | | | | WT rpc-1 + let-363/CeTOR<0.0001 | |
| | | | | | | | raga-1 control<0.05 | |
| | | | | | | | raga-1 rpc-1 dil.<0.05 | |
| | | | | | | | raga-1 ceTOR dil.<0.05 | |

All biological trials are indicated.

mutation did not achieve complete TORC1 reduction. However, this was non-additive with *rpc-1* RNAi (Fig 1G, *P* = 0.80). Together, our data suggest that CeTORC1 and Pol III act in the same pathway to affect organismal reproduction.

### TOR and RNA polymerase III interact genetically and temporally to control adult lifespan

TOR inhibition extends lifespan in a range of model organisms from yeast to mammals (27). In *C. elegans*, depletion of CeTOR signaling either by *raga-1/RagA* loss-of-function mutation or *let-363/CeTOR* RNAi extends lifespan (26, 28). Given that Pol III and TORC1 act epistatically in respect to progeny production (Fig 1E–G), we wanted to test the genetic interaction between CeTOR signaling and Pol III for *C. elegans* adult lifespan. As above, we reduced CeTOR signaling at two different time points: from embryonic stages using *raga-1/RagA* mutation and from the L4 stage using *let-363/CeTOR* RNAi, where it consistently initiating Pol III knockdown at the L4 stage (Fig 1C). We found that although *raga-1(ok386)* mutants are long-lived, the addition of *rpc-1* RNAi further extended this lifespan, indicating that CeTOR and Pol III are acting in parallel (Fig 2A, Table 1). To rule

out the possibility that *raga-1/RagA* mutation is incomplete and thus masking an epistatic relationship between TOR and Pol III, we examined the simultaneous knock down of *raga-1/RagA* with *let-363/CeTOR* with and without *rpc-1* RNAi (Fig 1A and C). In our hands, *raga-1* mutation and *let-363/CeTOR* RNAi were not additive; however, the addition of *rpc-1* RNAi did further increase the *raga-1/ RagA*; *let-363/CeTOR* lifespan (Fig 2B, Table 1). This supports our conclusion that CeTOR signaling engages additional, Pol III independent, mediators to control lifespan if knocked out during development (Fig 2C, Table 1).

In fruit flies, RNA Pol III acts downstream of TOR to regulate lifespan. RNA Pol III is activated when TORC1 signalling is inhibited with the use of rapamycin (18). In addition, the lifespan increases incurred by rapamycin treatment and Pol III knockdown (both from adulthood) are non-additive, indicating that Pol III mediates the effects of TOR for longevity (18). To test whether a similar scenario exists in worms, we knocked down CeTOR signaling only from the L4 stage using *let-363/ CeTOR* RNAi and examined its epistatic interaction with Pol III. Interestingly, although both knockdown of *let-363/CeTOR* or *rpc-1* from the L4 stage extended lifespan, there was no additive effect when the RNAi treatments were given simultaneously. This indicates that Pol III

mediates the effects of CeTOR signaling in adulthood to control lifespan in *C. elegans* (Fig 2C, Table 1). Taken together, our data suggest that the genetic interaction between CeTOR signaling and RNA Pol III is dependent on the timing of CeTOR reduction, and whereas reducing CeTOR signaling during development engages mediators in addition to Pol III to control lifespan, CeTOR reduction specifically in adulthood could act through Pol III to promote longevity.

### RNA polymerase III does not interact with global protein synthesis, insulin or germline signalling to control lifespan

Reducing CeTOR signaling during development engages mediators distinct from Pol III to affect lifespan (Fig 1C). These additional mediators may work downstream of CeTOR to drive organismal development. To identify these, we tested the genetic interaction of Pol III with candidate mediators implicated in protein synthesis or developmental processes. The ribosomal S6 Kinase (S6K) is a direct target of the TOR kinase and regulates global translation (29, 30). In *C. elegans*, *rsks-1* encodes for S6K and animals lacking this gene are long-lived (31) (Fig S3A, Table S1). We found that this longevity was additive, with that incurred by *rpc-1* RNAi treatment, suggesting that Pol III and S6K do not interact to extend lifespan (Fig S3A, Table S1). In mammals, another TOR target is the translational repressor, eukaryotic initiation factor 4E-binding protein (4eBP) (29, 32). *C. elegans* does not encode 4eBP, but animals with mutations in the translation initiation factors, *ife-2*/eIF4E and *ppp-1*/eIF2Bgamma, exhibit disruptions in protein translation (33). However, although deletion of either *ife-2* or *ppp-1* increased *C. elegans* lifespan, this was additive with *rpc-1* RNAi in both instances (Fig S3B and C, Table S1). In addition to CeTOR signalling, lifespan is also controlled by other signaling pathways implicated in development and growth (34). Mutation of the insulin receptor extends lifespan across species and has been implicated in protein synthesis and turnover in *C. elegans* (35, 36). In *C. elegans*, *daf-2* encodes the insulin receptor and *daf-2* loss-of-function mutants are long-lived (37) (Fig S4A, Table S2). However, knockdown of Pol III using *rpc-1* RNAi from the L4 stage was additive with *daf-2* lifespan, suggesting that Pol III-does not interact with insulin signalling for lifespan. Loss of germline signalling also consistently increases lifespan in model organisms (38). The *C. elegans* mutants *glp-1(e2141ts)* and *glp-4 (bn2)* are both sterile and live significantly longer than wild-types (39, 40). However, both of these mutations were additive with *rpc-1* RNAi (Fig S4B and C, Table S2) implying that Pol III-inhibition induced longevity is independent of germline signaling. Taken together, these data show that Pol III knockdown does not seem to affect the lifespan incurred by reducing global protein synthesis, although this does not rule out the possibility that Pol III affects protein synthesis by other mechanisms not interact with global translation mechanisms.

### RNA polymerase III knockdown acts in specific tissues to extend *C. elegans* lifespan

In both *C. elegans* and *D. melanogaster* the longevity phenotype caused by Pol III knockdown appears to be mediated in a tissue-specific manner (18). A detailed contribution of individual tissues has not been tested in worms, so to

characterize the expression pattern of Pol III in *C. elegans*, we examined the expression of a *rpc-1::gfp::3xflag* translational reporter in wild type adult animals (41). We found RPC-1::GFP in a wide variety of tissues including nuclei of muscle, intestine, hypodermis, neurons, and germline cells (Fig 3A). Treatment of this reporter strain with *rpc-1* RNAi, however, reduced expression of RPC-1::GFP in all of these tissues (Fig S5), indicating that these tissues could contribute to the Pol III-inhibition longevity phenotype.

To explore the requirement of these individual tissues for Pol III-knockdown mediated longevity, we used strains harboring a mutation in the dsRNA processing-pathway gene *rde-1*. The knockout mutant *rde-1(ne300)* is strongly resistant to dsRNA induced via the feeding RNAi method, with transgenic rescue of RDE-1 in different tissues allowing tissue-specific knockdown using RNAi (42). We knocked-down *rpc-1* using RNAi from early adulthood in the muscle, intestine, hypodermis, neurons and germline, and measured their lifespan. Interestingly, *rpc-1* RNAi, when knocked-down specifically in the body-wall muscles, extended the median lifespan by a modest 8%, thus indicating a role for this tissue in mediating Pol III knockdown longevity (Fig 3B, Table 2). However, we found that *rpc-1* RNAi in the hypodermis, germline, neurons or intestine could not increase lifespan compared with control (Fig 3C–F, Table 2). Taken together, this suggests that Pol III acts in *C. elegans* muscle to inhibit lifespan under normal conditions.

### Late-life knockdown of RNA polymerase III is sufficient to extend lifespan and improve age-related health

In mice, the inhibition of TOR by rapamycin has been shown to extend lifespan even when fed late in life (43). Given that the TOR-Pol III axis is important for adult lifespan (Fig 2C, Table 1), we asked whether Pol III knockdown in late adulthood could also extend longevity in *C. elegans*. To test this, we raised worms on *E. coli* OP50-1 until day 5 of adulthood and then transferred them to either control or *rpc-1* RNAi plates. Excitingly, compared with control fed animals, worms on *rpc-1* RNAi lived longer, suggesting that Pol III knockdown in late adulthood effectively extends longevity (Fig 4A, Table 3). Indeed, this late-life knockdown of Pol III resulted in a lifespan almost identical to that incurred by L4 RNAi treatment (Fig S6, Table 3, Filer et al, 2017 (18)). Overall, these results show that Pol III knockdown in late adulthood is sufficient to extend *C. elegans* lifespan, strengthening our earlier finding that the beneficial longevity effects of Pol III are post-reproduction.

Healthspan is an important indicator of youthfulness and virility and, ideally, an increase in lifespan is complemented by improvement in organismal healthspan (44). We investigated whether Pol III knockdown could also enhance the health and fitness of worms across their lives by testing their ability to move in liquid media using a thrashing assay. As expected, thrashing ability decreased with age but compared with controls, worms on *rpc-1* RNAi moved better at later time points (day 7 and 11 of adulthood) than controls, suggesting improved fitness and better late-life health (Fig 4B). We then compared the ability of *C. elegans* with move independently on food over their lifespan by videoing their natural movement on *E. coli* seeded plates. Similarly to the thrashing assay,

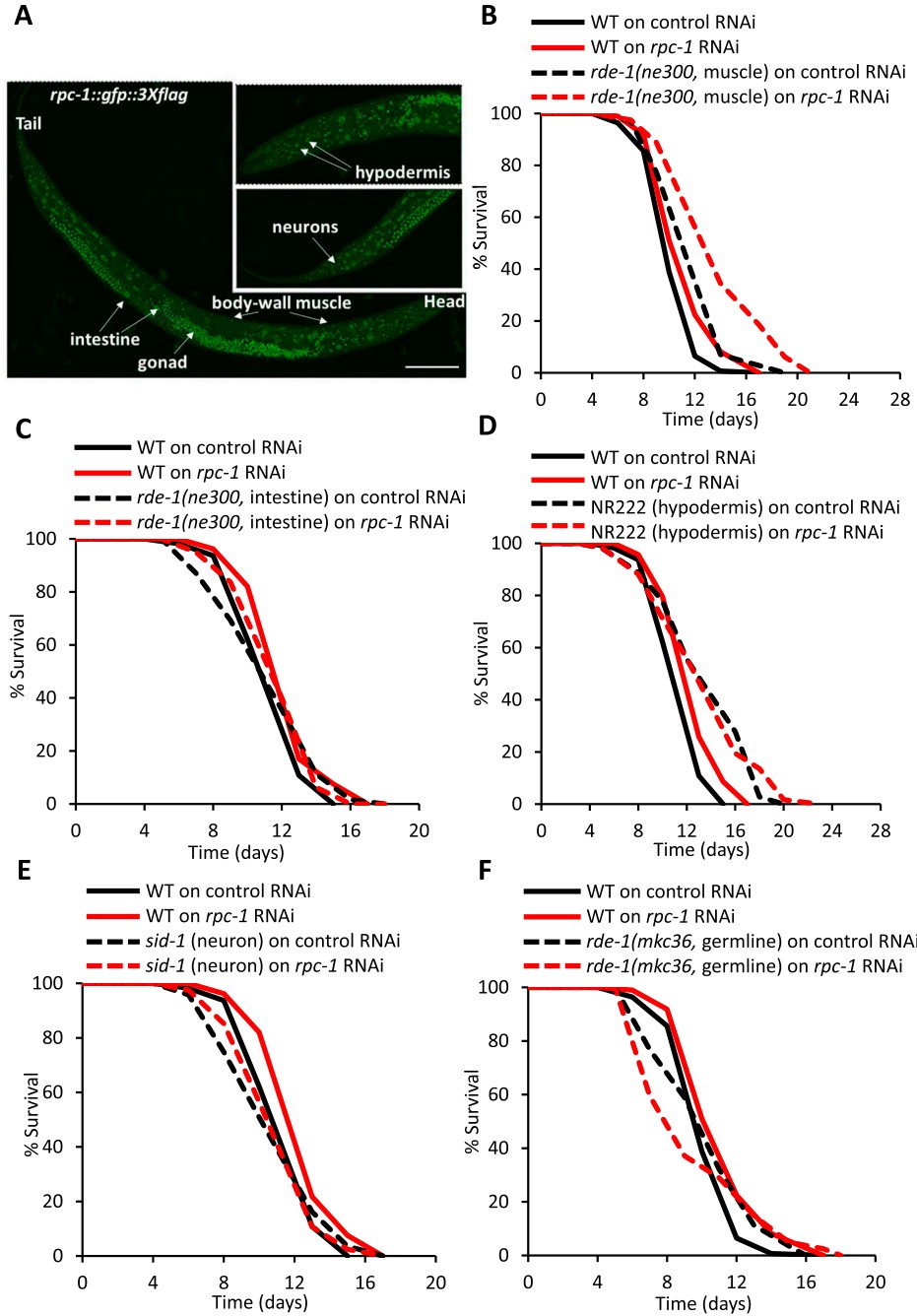

Figure 3. RPC-1 acts tissue specifically to control lifespan.
**(A)** RPC-1::GFP expression in L4 *C. elegans*. In each labelled tissue, RPC-1::GFP is localized in the nuclei. Representative images shown. We also noted similar expression patterns throughout adulthood (see also Fig S5). Scale bar indicates 50 *μm*. **(B, C, D, E, F)** *rpc-1* RNAi specifically in body-wall muscle extends lifespan, but *rpc-1* knockdown in the intestine, hypodermis, neurons or germline does not extend lifespan. The intestine-specific, hypodermis-specific, germline-specific and muscle-specific RNAi rescue *rde-1* in the *rde-1(ne300)* strain is under control of the *mtl-2, col-62, sun-1,* and *myo-3* promoters, respectively. **(B, C, D, E, F)** One representative experiment is shown, refer to Table 2 for data on all replicates and statistical analysis.

*C. elegans* moved around on plates less as they aged, but older worms treated with *rpc-1* RNAi moved better compared with controls (Fig 4C, day 5 adults *P* = 0.0003, day 8 adults *P* = 0.002 compared with age-matched controls). Together with the thrashing data, this supports a role for Pol III in mediating life-long health and overall lifespan.

Given that Pol III knockdown in muscle is sufficient to promote longevity, we also tested whether muscle-specific *rpc-1* RNAi improved age-related fitness. We found that indeed *rpc-1* knockdown in the muscle was sufficient to improve overall body movement on day 5 of adulthood (Fig 4D, *P* = 0.001 compared with control). This supports our finding that *rpc-1* is acting on muscles to control age-related health.

It is well reported that ageing contributes to disorganization of the sarcomeres in the body-wall muscle, a phenomenon called sarcopenia which is also a general hallmark of mammalian ageing (45, 46). It has also been shown that changes in muscle mitochondrial structure are strongly correlated with the decline of both sarcomeric structure and speed of movement (47). To further explore the underlying dynamics that might contribute to the improved muscle

**Table 2. Lifespan data for indicated figures including all biological trials.**

| Figure | Trial | Strain | Genotype | Tissue affected by RNAi | Mean lifespan (days) | P-value (Log-rank) vs | N dead |
|---|---|---|---|---|---|---|---|
| 3B | 1 | WT | control RNAi | Body-wall muscle | 10.32 | | 139 |
| | | WT | rpc-1 RNAi | | 11.3 | WT cont.<0.05 | 98 |
| | | WM118 | control RNAi | | 9.12 | | 93 |
| | | WM118 | rpc-1 RNAi | | 9.93 | WM118:Cont<0.05 | 100 |
| | 2 | WT | control RNAi | Body-wall muscle | 10.11 | | 78 |
| | | WT | rpc-1 RNAi | | 10.98 | WT cont.<0.05 | 99 |
| | | WM118 | control RNAi | | 10.13 | | 101 |
| | | WM118 | rpc-1 RNAi | | 9.99 | WM118:Cont<0.05 | 84 |
| | 3 | WT | control RNAi | Body-wall muscle | 10.486 | | 88 |
| | | WT | rpc-1 RNAi | | 11.589 | WT cont.<0.05 | 68 |
| | | WM118 | control RNAi | | 10.22 | | 66 |
| | | WM118 | rpc-1 RNAi | | 11.01 | WM118:Cont<0.05 | 78 |
| 3C | 1 | WT | control RNAi | Intestine | 10.51 | | 111 |
| | | WT | rpc-1 RNAi | | 11.48 | WT cont.<0.05 | 106 |
| | | IG1839 | control RNAi | | 10.72 | | 62 |
| | | IG1839 | rpc-1 RNAi | | 11.01 | IG1839:Cont RNAi:NS | 80 |
| | 2 | WT | control RNAi | Intestine | 10.77 | | 68 |
| | | WT | rpc-1 RNAi | | 11.85 | WT cont.<0.05 | 52 |
| | | IG1839 | control RNAi | | 10.33 | | 72 |
| | | IG1839 | rpc-1 RNAi | | 10.39 | IG1839:Cont RNAi:NS | 79 |
| | 3 | WT | control RNAi | Intestine | 10.486 | | 88 |
| | | WT | rpc-1 RNAi | | 11.389 | WT cont.<0.05 | 68 |
| | | IG1839 | control RNAi | | 11.692 | | 92 |
| | | IG1839 | rpc-1 RNAi | | 11.395 | IG1839:Cont RNAi:NS | 78 |
| 3D | 1 | WT | control RNAi | Hypodermis | 10.71 | | 111 |
| | | WT | rpc-1 RNAi | | 11.65 | WT cont.<0.05 | 93 |
| | | NR222 | control RNAi | | 12.8 | | 108 |
| | | NR222 | rpc-1 RNAi | | 12.65 | NR222:Cont RNAi :NS | 118 |
| | 2 | WT | control RNAi | Hypodermis | 10.71 | | 111 |
| | | WT | rpc-1 RNAi | | 11.65 | WT cont.<0.05 | 106 |
| | | NR222 | control RNAi | | 12.1 | | 83 |
| | | NR222 | rpc-1 RNAi | | 12.08 | NR222:Cont RNAi :NS | 79 |
| | 3 | WT | control RNAi | Hypodermis | 10.56 | | 72 |
| | | WT | rpc-1 RNAi | | 11.37 | WT cont.<0.05 | 56 |
| | | NR222 | control RNAi | | 12.78 | | 90 |
| | | NR222 | rpc-1 RNAi | | 12.1 | NR222:Cont RNAi :NS | 94 |
| 3E | 1 | WT | control RNAi | Neurons | 10.71 | | 111 |
| | | WT | rpc-1 RNAi | | 11.59 | WT cont.<0.05 | 106 |
| | | TU3401 | control RNAi | | 10.08 | | 167 |
| | | TU3401 | rpc-1 RNAi | | 10.43 | TU3401:Cont RNAi:NS | 122 |
| | 2 | WT | control RNAi | Neurons | 10.11 | | 91 |
| | | WT | rpc-1 RNAi | | 11.25 | WT cont.<0.05 | 77 |
| | | TU3401 | control RNAi | | 9.98 | | 78 |
| | | TU3401 | rpc-1 RNAi | | 9.90 | TU3401 :Cont RNAi:NS | 78 |

**Table 2. Continued**

| Figure | Trial | Strain | Genotype | Tissue affected by RNAi | Mean lifespan (days) | P-value (Log-rank) vs | N dead |
|---|---|---|---|---|---|---|---|
| 3F | 1 | WT | control RNAi | Germline | 10.57 | | 139 |
| | | WT | *rpc-1 RNAi* | | 11.53 | WT cont.<0.05 | 98 |
| | | DCL569 | control RNAi | | 10.62 | | 168 |
| | | DCL569 | *rpc-1 RNAi* | | 9.93 | DCL569:Cont RNAi:NS | 121 |
| | 2 | WT | control RNAi | Germline | 10.57 | | 111 |
| | | WT | *rpc-1 RNAi* | | 11.53 | WT cont.<0.05 | 106 |
| | | DCL569 | control RNAi | | 9.93 | | 99 |
| | | DCL569 | *rpc-1 RNAi* | | 10.12 | DCL569:Cont RNAi:NS | 101 |
| S8 | 1 | WT | control RNAi | Intestine | 10.98 | | 88 |
| | | WT | *rpc-1 RNAi* | | 12.12 | WT cont.<0.05 | 72 |
| | | VP303 | control RNAi | | 9.76 | | 82 |
| | | VP303 | *rpc-1 RNAi* | | 8.89 | VP303:Cont RNAi<0.05 | 69 |
| | 2 | WT | control RNAi | Intestine | 10.11 | | 91 |
| | | WT | *rpc-1 RNAi* | | 11.25 | WT cont.<0.05 | 77 |
| | | VP303 | control RNAi | | 9.90 | | 71 |
| | | VP303 | *rpc-1 RNAi* | | 10.98 | VP303:Cont RNAi<0.05 | 78 |

function in *rpc-1* knockdown animals, we examined the mitochondrial structures using a marker of body-wall muscle mitochondria. Interestingly, knock down of *rpc-1* improved the organization of mitochondrial networks in older worms compared with control (Fig 4E *P* = 0.033). Taken together, these results suggest that knockdown of Pol III in adulthood leads to significant increase in lifespan because of improvement in muscle-specific health parameters.

# Discussion

### RNA polymerase III and TOR signalling interact temporally to control development and lifespan

Both TOR signalling and RNA Polymerase III are important regulators of anabolic processes, and control a wide array of cellular functions including reproduction and lifespan (16). Here we show that TOR and Pol III interact in a time-dependent manner, and that knockdown of each component individually, sequentially or in parallel influences the outcome of both reproductive and longevity processes (Fig 5). In *C. elegans*, we manipulated CeTOR signalling using both genetic mutants (that constitutively reduce CeTOR signalling throughout life) and *let-363/CeTOR* RNAi (which reduces CeTOR signalling the late L4 stage). We show that the genetic relationship between CeTOR signaling and Pol III is timing dependent. The requirement and collaboration of TOR and Pol III to support developmental processes, and the fact that their reduction in adulthood extends lifespan supports the antagonistic pleiotropy theory of ageing; whereby, certain genes are switched "on" to promote the anabolic programs leading to growth, sexual maturity and reproductive fitness, and "hyperfunction" of these genes in adulthood leads to senescence and ageing (48).

The embryonic lethality of *rpc-1* and *let-363/CeTOR* meant that these genes must be knocked down specifically in adulthood (49, 50).

Thus, our epigenetic approach meant that artifacts or reduced RNAi efficacy relating to use of combinatorial RNAi could arise (51). However, we observe the same fecundity and lifespan phenotypes for each gene whether it was knocked down individually or diluted 1:1 with control RNAi (Fig S2A–C and Table 1). In fact, these phenotypes were striking, both supporting our methodology and demonstrating the sensitivity of *C. elegans* to RNAi knockdown of these crucial genes.

The contrasting effects on fitness and longevity as observed by knockdown of Pol III in early development as opposed to late adulthood are similar to those observed for other signaling pathways including mTOR and insulin. Knockdown of *daf-2* from early to mid-adulthood also promotes longevity whereas its knockdown from early larval stages reduces fecundity and reproductive fitness (52). Late-life interventions targeting IGF-1 receptor have also been shown to improve lifespan and healthspan in mice (53). However, whereas the lifespan of *daf-2* worms cannot be further extended by *let-363/CeTOR* inhibition, suggesting a common mechanistic pathway (28), we did not find evidence for a genetic interaction with the *C. elegans* insulin receptor *daf-2* (Fig S4A). In fact, *daf-2* mutation and *rpc-1* RNAi were additives in our hands, implying independent downstream effectors of Pol III and DAF-2. Our similar results with germline-signaling mutants (Fig S4B and C), which increase longevity by reducing germline activity, lead us to speculate that this could be in part due to the common transcriptional denominator DAF-16, which integrates signals both from the IGF-1/DAF-2 axis and germline-loss driven longevity (38, 54, 55), whereas Pol III knockdown bypasses DAF-16 requirements entirely.

### RNA polymerase III inhibition as a modulator of late-life health and lifespan

Studies in a wide variety of model organisms show that mTOR inhibition increases longevity and improves organismal health,

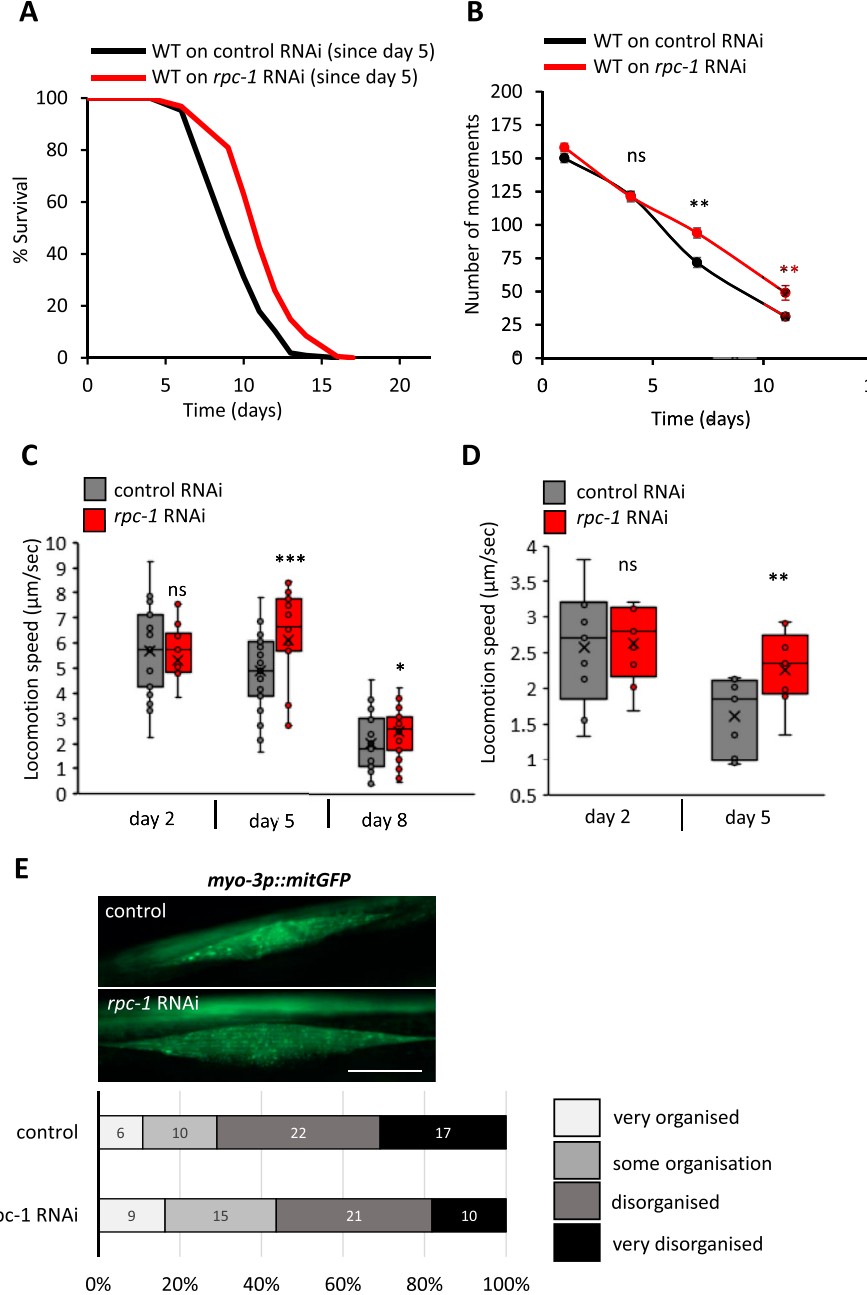

**Figure 4. Reducing RNA polymerase III in late life extends lifespan and improves health.**
**(A)** *rpc-1* RNAi in late-adulthood extends lifespan. One representative experiment is shown, refer to Table 3 for data on all replicates and statistical analysis. **(B)** *rpc-1* RNAi improves *C. elegans* thrashing ability in liquid as they age. Measurements taken on days 1, 4, 7, and 11 of adulthood. Combined data from three trials shown, n = < 60 worms per group. *t* test against age-matched control: **$P < 0.005$; ns, non-significant. **(C)** *rpc-1* RNAi improves normal *C. elegans* movement in late life. **(D)** *rpc-1* RNAi specifically in muscle improves normal *C. elegans* movement in late life. Note that the strain used for muscle specific RNAi also carries a *rol-6* marker; therefore, the average movement is decreased compared with those exhibiting normal sinusoidal movement. **(E)** *rpc-1* RNAi improves mitochondrial network organization in body-wall muscles. *N* = <55 per condition over three independent experiments. Scoring system is shown in Fig S7. Scale bar indicates 20 *μm*. In (C, D) One representative experiment is shown. n = <20 worms per group. Statistical comparisons were made by One-way ANOVA. *P*-value: ***$P < 0.0001$; **$P < 0.005$; *$P < 0.05$; ns, non-significant. In (E) control versus *rpc-1*, RNAi was compared using a Chi-square test $P = 0.033$.

spurring interest in mTOR inhibitors for slowing down human ageing (56). However, this can occur at the cost of development, e.g., treating female flies with rapamycin increased lifespan, but significantly reduced brood size (57, 58). In addition, clinical studies have shown that chronic exposure to mTOR inhibitors can result in serious side effects including immunosuppression and metabolic dysfunction, making them unattractive anti-ageing therapies (27). Consequently, transient knockdown of mTOR genetically, or by inhibitors, is a fast emerging paradigm for targeting mTOR complexes. Studies targeting mTORC1 by rapamycin even late in life have shown to extend lifespan in mice models, without major side effects (43, 59). Here we show that whereas Pol III knockdown during early larval stages leads to developmental abnormalities and larval arrest (Fig 1B), Pol III knockdown post-reproduction on the fifth day of adulthood, extends lifespan, suggesting that, similar to TOR, even late perturbations in Pol III activity are sufficient to promote longevity (Fig 4A). Importantly, we also show that Pol III inhibition is sufficient to improve age-related decline in movement (Fig 4C and D), which has consistently been correlated with fitness and improved lifespan in worms (60, 61, 62). Overall, our results make RNA Pol III a promising target molecule for ameliorating post-reproductive age-related decline.

**Table 3. Lifespan data for indicated figures including all biological trials.**

| Figure | Trial | Strain | Genotype | Mean lifespan (days) | *P*-value (Log-rank) vs | *N* dead |
|--------|-------|--------|----------|----------------------|-------------------------|----------|
| 4A | 1 | WT | control RNAi (since day 5) | 9.95 | | 106 |
| | | WT | *rpc-1* RNAi (since day 5) | 11.35 | WT cont.<0.0005 | 189 |
| | 2 | WT | control RNAi (since day 5) | 10.02 | | 89 |
| | | WT | *rpc-1* RNAi (since day 5) | 11.15 | WT cont.<0.005 | 91 |
| | 3 | WT | control RNAi (since day 5) | 9.13 | | 92 |
| | | WT | *rpc-1* RNAi (since day 5) | 11.20 | WT cont.<0.005 | 86 |
| S6 | 1 | WT | control RNAi (since L4) | 9.89 | | 59 |
| | | WT | *rpc-1* RNAi (since L4) | 11.23 | WT cont.<0.005 | 59 |
| | 2 | WT | control RNAi (since L4) | 9.97 | | 62 |
| | | WT | *rpc-1* RNAi (since L4) | 11.89 | WT cont.<0.005 | 66 |

### RNA polymerase III acts in muscle to drive ageing

It is known that aging progresses heterogeneously and various tissue-specific responses have been identified to perturb organismal aging. Earlier studies have shown that knockdown of Pol III in the fly gut improves gut barrier function and is sufficient to ameliorate age-related impairments in gut health which may ultimately promote longevity (18). In *C. elegans*, multiple studies have shown tissue-specific requirements for longevity genes (55, 63) and Pol III RNAi mediated longevity can also be mediated via the intestine (18). The broad expression of *rpc-1::GFP* (Figs 3A and S5), led us to explore its longevity and health functions in other tissues and found that muscle-specific knockdown of *rpc-1* was sufficient to extend lifespan (Fig 3B).

Sarcopenia, which leads to debilitating conditions among the elderly, is characterised by low skeletal muscle mass or strength and is believed to be the most frequent cause of disability and a major risk factor of health-related conditions (64, 65). In *C. elegans*, even reducing translation in body-wall muscle during development shortens lifespan and increases reproduction (66). Conversely, multiple studies, in both worms and mammals, have shown that improving body-wall muscle function can improve overall health and longevity (67, 68, 69), with muscle mitochondrial organisation being a critical contributing factor. For example, loss of RAGA-1/RagA in the nervous system increases lifespan via maintaining mitochondrial networks in muscle (70). Here we show that reducing Pol III/*rpc-1* preserves both mitochondrial organisation (Fig 4E) and suppresses the age-related decline observed in older worms (Figs 4D and S7). This strongly supports a role for Pol III in maintaining muscle function with age, with the potential to be used as a novel target molecule for interventions aimed at reducing sarcopenia in human ageing.

Previously, we showed that *rpc-1* RNAi in the intestine-specific RNAi strain VP303 significantly extends lifespan as reported in Filer et al, 2017 (18) (Fig S8). This observation is in contrast to our published data wherein *rpc-1* RNAi extends the lifespan of a different intestine-specific RNAi strain *rde-1(ne219)*. However, this difference is likely attributed to the allele-specific differences between *rde-1(ne219)* and *rde-1(ne300)*; *ne219* retains significant RNAi processing capacity and is not completely null; whereas, *ne300* deletion allele is completely resistant to RNAi (Watts et al (42)). Thus, the *rde-1(ne300)* strain we initially tested is likely to be "leaky" and RNAi knockdown might not be limited to the gut of the worms.

Lowering translation or mTORC1 activity in adulthood consistently increases lifespan (20), reduces brood size and improves motility (31, 71). Surprisingly though, suppressing CeTORC1 specifically in body-wall muscle increased reproduction and slowed motility, indicating a role for the muscle in CeTOR-mediated activity (72). We could not test if muscle-specific reduction in Pol III during development altered lifespan owing to the strong larval arrest phenotype of *rpc-1* RNAi but suspect that such an intervention may perturb global protein translation and remodel inter-tissue communication via one or many of the small-RNAs transcribed by Pol III.

### Exploring additional mediators of the Pol III response

It has been well documented that inhibition of translation delays ageing and evidence suggests that reducing global translation can maintain protein homeostasis, dysregulation of which leads to aging and senescence (71, 73, 74). It is likely that Pol III reduction leads to a decrease in two major RNA types, tRNAs and 5s rRNA, subsequently down-regulating global protein translation. Hence, we were surprised to find that *rpc-1* RNAi induced longevity was additive with *C. elegans* mutants of other key translation mediators (Fig S3). This implies a more nuanced role of Pol III in maintaining a healthy proteome, possibly via improving protein folding or reducing insoluble protein loads, which would be important for lifespan and age-related disease. In the future, addressing the impact of Pol III inhibition on protein levels and quality should prove interesting.

## Materials and Methods

### *C. elegans* maintenance and strains

*C. elegans* were routinely grown and maintained on Nematode Growth Media (NGM) seeded with *E. coli* OP50-1. The wild-type strain was Bristol N2. The following strains were used:VC222 *raga-1(ok386)*,RB1206 *rsks-1(ok1255)*, RB579 *ife-2 (ok306)*, *ppp-1(syb7781)*,

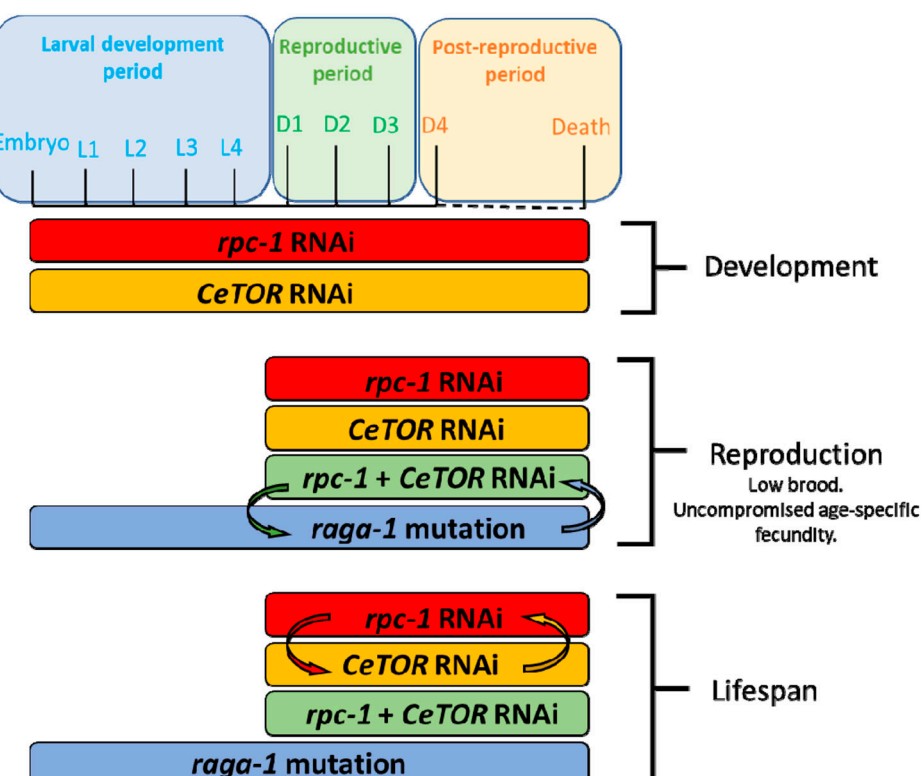

**Figure 5. Temporal genetic interactions between TORC1 and RNA Polymerase III.** Model summarizing the interactions between CeTORC1 and RNA Polymerase III required for development, reproduction and lifespan. This relationship is highly temporal: Reducing CeTOR signalling specifically during adulthood requires RNA Pol III for reproduction and lifespan, whereas CeTORC1 reduction during the embryonic and larval period requires other mediators to promote reproduction and longevity. Data supporting this model are shown in Figs 1 and 2 and S1 and S2.

DR1567 *daf-2*, CB4037 *glp-1(e2141)*, SJ4103 *myo-3p::mitGFP*, SS104 *glp-4(bn2)*, IG1839 *frSi17 II; frIs7 IV; rde-1(ne300) V* (intestine-specific RNAi), DCL569 *mkcSi13 II; rde-1(mkc36) V* (germline-specific RNAi). WM118 *rde-1(ne300) V; neIs9 X* (muscle-specific RNAi). TU3401 *sid-1(pk3321) V; uIs69 V* (neuronal-specific RNAi), NR222 *rde-1(ne219) V; kzIs9* (hypodermis-specific RNAi), VP303 *rde-1(ne219) V; kbIs7* (intestine-specific RNAi). NB: The *raga-1(ok386)* allele harbors a 1,242 bp deletion at the *raga-1/RagA* locus that removes almost the entire coding region of the gene.

### Lifespan assay

Longevity assays were performed as previously described (Filer et al, 2017 (18)). Briefly, synchronous L1 animals were placed on NGM plates seeded with *E. coli* OP50-1 until they reached L4. From L1 to L4 larval stage, animals were maintained at 20°C. At L4 stage, worms were shifted to plates seeded with *E. coli* HT1115 carrying appropriate RNAi clones. These RNAi clones were obtained from the Ahringer RNAi library. The plates were supplemented with 50 µM FuDR solution and lifespans were carried out at 25°C. For combinatorial RNAi knockdown, the two respective overnight RNAi cultures were mixed 1:1 before seeding the plates. In these experiments, controls were also diluted 1:1 with HT115 expressing empty pL4440. Live, dead and censored worms were counted every 2–3 d in the worm populations by scoring movement with gentle prodding when necessary. Data were analyzed and statistics performed using the online freeware OASIS (http://sbi.postech.ac.kr/oasis/surv/).

### Brood size measurements

Animals were grown on *E. coli* OP50-1 plates at 20°C till L4 larval stage. At L4, ~15 animals were placed on individual NGM plates seeded with the appropriate RNAi. For combinatorial RNAi, see protocol in lifespan section. Progeny production was assessed at 25°C and each parent animal moved to a fresh plate daily until egg-laying ceased. The total number of eggs and larvae were counted after 24 h. Plates in which the animals went missing or showed intestinal bursting were not included in the final analysis. NB: At 25°C (the temperature that we did these experiments) very few animals suffered internal hatching and those that did were taken out of the analysis.

### Thrashing assay

Animals were raised to L4 for lifespan assays. At L4, the animals were moved to control RNAi or *rpc-1* RNAi seeded plates. On days 4, 7, and 11 of adulthood, worms were put in the isotonic M9 buffer and allowed to acclimatize for 1 min before measuring the number of side-to-side mid-body movements per minute. Fifteen animals were analyzed per condition.

### Microscopy

*C. elegans* carrying the *rpc-1::gfp::3Xflag* reporter were grown in fed conditions to the L4 stage. *myo-3::mitGFP* imaging, animals were imaged on day fifth of adulthood with *rpc-1* RNAi treatment initiated on day 1. When imaging, animals were immobilized on 2% agarose

pads using 0.06% Levamisole and imaged immediately using a Zeiss Confocal Ultra at 40X zoom.

## Speed measurement

*C. elegans* were raised and maintained as for the lifespan assays. At each timepoints, 20–30 worms were moved to a thinly seeded 60 mm plate and were allowed to acclimatize for 10 min. Their individual movements were then recorded using Wormtracker (MBF Bioscience) for 2 min with 60 fpm settings. Average speeds for individual animals were calculated using MS Excel.

## Spermatocyte measurement

Hermaphrodite *C. elegans* were raised and maintained for brood size assays. At the early L4 stage, 30–40 worms were moved to RNAi plates. After 24 h (day 1 of adulthood), worms were washed with 0.01% Tween PBS to remove excess bacteria, fixed in methanol for 5 min. Worms were washed once again with 0.1% Tween PBS. DAPI dye was added immediately and kept for 10 min. Worms were washed once again with 0.1% Tween PBS and mounted on 2.5% agarose pads for visualization. DIC and DAPI images were acquired using a Olympus IX 81 microscope. Individually, distinct spermatocytes for each worm were counted.

# Data Availability

All relevant data are within the manuscript and its Supporting Information files.

# Supplementary Information

# Acknowledgements

This work was funded by a BBSRC NI award BB/R003629/1 to JMA Tullet and BBSRC award BB/S014365/1 to JMA Tullet, N Alic and C Selman. Some strains were obtained from *Caenorhabditis* Genetics Center, Minnesota, USA, which is funded by the NIH Office of Research Infrastructure Programs (P40 OD01044). *ppp-1(syb7781)* was a kind gift from the Denzel laboratory (Cologne, Germany).

## Author Contributions

Y Malik: investigation and writing—original draft.
I Goncalves Silva: investigation.
RR Diazgranados: investigation.
C Selman: formal analysis, funding acquisition, and writing—original draft.
N Alic: formal analysis, funding acquisition, and writing—original draft.
JMA Tullet: conceptualization, supervision, methodology, project administration, and writing—original draft, review, and editing.

## Conflict of Interest Statement

The authors declare that they have no conflict of interest.

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
