## [Reviewer comments · Life Science Alliance]

Life Science Alliance

Timing of TORC1 inhibition dictates Pol III involvement in *Caenorhabditis elegans* longevity

Yasir Malik, Isabel Goncalves Silva, Rene Diazgranados, Colin Selman, Nazif Alic, and Jennifer Tullet

DOI: <https://doi.org/10.26508/lsa.202402735>

Corresponding author(s): Jennifer Tullet, University of Kent

Review Timeline:

Submission Date:	2024-03-25
Editorial Decision:	2024-04-15
Revision Received:	2024-04-25
Accepted:	2024-04-26

Transaction Report:

Please note that the manuscript was reviewed at *Review Commons* and these reports were taken into account in the decision-making process at *Life Science Alliance*.

Reviews

Review #1

The manuscript reports effects on brood size, lifespan and healthspan upon manipulation of *C. elegans* genes encoding RagA, TOR and Pol III orthologs, as well as other well-characterized lifespan-affecting genes. The results point to complex relationships among TOR and Pol III that are not fully resolved, suggest a role for *rpc-1* Pol III that is additive with well-characterized lifespan pathways, indicate a late-life requirement for *rpc-1* Pol III to limit lifespan, and, contrary to a previous publication, suggest a muscle requirement for *rpc-1* Pol III for lifespan limitation.

****Major comments**** regarding key conclusions:

- 1) The work demonstrates that brood size is reduced upon *rpc-1* Pol III RNAi feeding from the L4 stage. However, no further analysis is provided to show how later aspects of reproduction impair brood. Minimally, ruling out effects on spermatogenesis would be important since sperm number limits self-fertile brood size. It is also unclear from the methods whether the brood size results include embryonic lethality (post-reproduction). Internal hatching, if it occurred, could also affect interpretation of the results. A change in the reproductive period should be noted if it occurred.
- 2) The authors claim that, similar to the relationship previously concluded from aging studies, *rpc-1* acts downstream of TORC1. However, this claim is not well supported. In an effort to circumvent early lethality caused by loss of *let-363* ("CeTOR"), they use a mutation in *raga-1* RagA and demonstrate a further reduction in brood with *rpc-1* RNAi. If *raga-1(ok386)* were a null this result would demonstrate a relationship that is at least partially parallel, not linear. By contrast, double RNAi with *let-363* was "non-additive", suggesting a more linear relationship. However, interpretation of these experiments requires (1) that the *raga-1* mutation is null and affects only TORC1 signaling, (2) evidence that the double RNAi worked well (e.g., qPCR; see Ahringer et al. 2006 review regarding issues with multi-RNAi), and (3) failure to consider alternative effects of loss of *let-363* (e.g., TORC2). Negative results with RNAi are particularly problematic in the absence of convincing evidence that the RNAi worked well. Moreover, results in Figure 1G are difficult to interpret since the initial values are low. Here and elsewhere the genetics descriptions are unconventional, hampering interpretation. For example, what is meant by a mutation being "incomplete"? That it acts as a hypomorph?
- 3) Another claim is that *rpc-1* Pol III limits adult lifespan downstream of TOR. These results are not convincing. The two treatments (*raga-1* mutation as "embryonic" and L4 stage "CeTOR" *let-363* RNAi as late) are not directly comparable for reasons noted above, and the double RNAi problem hampers interpretation. The nomenclature might be easier to follow if the authors state the actual *C. elegans* genes manipulated (e.g., *let-363* TOR versus *raga-1* RagA) rather than using "CeTOR" as a catch-all since these genes are not identical in action.
- 4) Based on genetic interactions (*rsk-1*, *ife-2*, *ppp-1*, *daf-2* and germline loss) they show that *rpc-1* RNAi further extends the long lifespan conferred by each of the mutant alleles tested, as well as germline loss induced by two different mutant conditions. These results, though negative, are important. The statement that *rpc-1* does not affect global protein synthesis is somewhat overstated without additional experimental support.
- 5) Extending and challenging their own previous work showing an intestinal focus of activity for *rpc-1* in limiting longevity (Filer et al., 2017), and noting that *RPC-1::GFP* detection can be knocked down by RNAi in several tissues, they use a tissue restricted *rde-1* expression approach (or *sid-1* for neurons) to test the contribution of intestine, hypodermis, neurons, muscle and germline. This new analysis points to a role for the muscle. This result is intriguing and warrants further experiments. To shore up tissue-specific claims the authors could consider (1) additional drivers for intestine and muscle *rde-1* in the RNAi experiments, or, ideally, a different approach such as tissue-specific protein degradation (again with multiple drivers), (2) a sufficiency experiment for muscle (wild-type muscle expression in the mutant to demonstrate reversal of the phenotype, or rescue of RNAi defects with an RNAi-insensitive reagent expressed in muscle). The possible explanation for the differences in *rde-1* results from the previous work should not be buried in the legends of Figure 3 and Figure S3. Perhaps this leaky background hypothesis should be directly tested (e.g., using the *RPC-1::GFP* to examine

whether residual expression exists in ne219 but not in ne300)? In any case, legend to Figure S3 needs editing: The ne219 background is not itself "intestine-specific", as implied, and the last sentence of Figure S3 legend should be "Thus, the rde-1(ne219)...", right?

6) Finally, they show that late-adult rpc-1 RNAi extends lifespan over control RNAi and that, by several movement assays, healthspan is improved upon L4 rpc-1 RNAi, even when RNAi is active in muscle (based on WM118).

7) The most significant new results are that rpc-1(RNAi) affects brood size, can extend lifespan (though modestly) after day 5 of adulthood, and that muscle may be involved rather than intestine.

****Additional comments****

Text throughout should clarify TOR vs presumed TORC1. Methods are insufficient. Important aspects of the lifespan methods and raw data are missing - e.g. exact numbers of worms censured. Exact information regarding statistical analysis is lacking (e.g., which tests, corrections for multiple testing). References should be given for all strains. For the rde-1 strains, it would be helpful to include, in addition to the transgene alleles, the actual promoters used to claim tissue specificity. Note, worms do not have "skeletal" muscle, as implied in the discussion. Figure 5 was not helpful for this reviewer. Figure legend to S3A is confusing: the intestinal signal appears stronger or at least equal, not weaker, in the rpc-1 RNAi background. Were these images collected using the exact same exposure settings?

See above. Study will be of interest to aging community.

Review #2

The study by Malik and Silva et al describes results of the study investigating the role of RNA Polymerase III in regulating fecundity and lifespan in *C. elegans*. The authors show that knockdown of Pol III, similar to mTOR suppression, is detrimental for reproduction. Likewise, suppression of either Pol III or mTOR in adult animals extends lifespan via apparently the same pathway. In contrast, Pol III knockdown has an additive effect on lifespan in combination with other established genetic lifespan-extending approaches suggesting that they are working via different mechanisms. Furthermore, using the tissue-specific knockdown of Pol III the authors found that suppression Pol III expression in the muscle, but not other major worm tissues, is sufficient for its lifespan extending effect. Finally, the lifespan extension is also observed when Pol III knockdown is initiated late in adulthood. The overall conclusion is that suppression of Pol III expression late in animal life, particularly in the muscle, is a potential strategy to extend life- and health-span. Overall, the study is well-designed, the tools and results are robust and analysed appropriately. The data presentation is excellent, and the manuscript is clearly written. Addressing the points below will help to improve the clarity further.

****Major:****

Significant amount of GFP signal is still present in RNAi treated animals, what is the tissue that maintains particularly high levels of expression (Fig. 3A) and how does it affect the conclusions?

What is the level of Pol III reduction in different tissues? Could different efficiency of knockdown explain the tissue-specific effect of Pol III downregulation on lifespan? It would be important to show (and, if possible, to quantify) the knockdown efficiency in different tissues using the available reporter.

****Minor:****

Fig. S3B is not cited in the text and the legend for the figure is somewhat confusing, potentially containing errors, this needs to be clarified.

This is the first thorough study of Pol III knockdown as a lifespan extending strategy in *C. elegans*. In addition to the different laboratory model (previous study of Pol III in ageing primarily focused on *Drosophila*), this

manuscript also offers several novel insights into consequences of Pol III perturbation at phenotypic, as well as mechanistic level in terms of interaction with other longevity pathways. The study will be of interest to those interested in processes underlying longevity and ageing. Considering that this topic is currently in fashion the publication will probably attract attention of not only specialist but also general public.

My expertise is in cellular proteostasis and its perturbation in age-related diseases.

Review #3

****Summary:****

The paper by Yasir Malik et al investigates the genetic interrelationship between TOR signalling and Pol III expression regarding fecundity and longevity in *C. elegans*. Based on a previous study that defined a role of Pol III downstream of TOR in longevity across various species, this study looks particularly at the relative timing and tissue requirements for TOR and Pol III inhibition in longevity. Data indicate that Pol III acts downstream of TOR in regulating fecundity while there are additive effects regarding survival. The Pol III effect on longevity is based on its role in the muscle. Finally, health-span parameters mirror the survival data.

****Major comments:****

This is a nice study that relies on genetic interaction to ask how TOR and Pol III interact. I find the observation that Pol III inhibition extends survival when initiated at day 5 of adulthood very exciting. In general, the study would benefit from additional data that back up the genetic observations.

1. In Fig. 1, experiments are done to inhibit TOR to varying degrees in order to perform epistasis experiment. Of course these are difficult to interpret without the use of full KOs/loss of function. So while this is a good solution, it would be important to quantify the level to which TOR signalling is inhibited, optimally with biochemical experiments.

2. General brood size is very low in the WT worms. Normally, one would expect 250-300 offspring per adult worm. It would be helpful if the authors could address this.

3. Why were lifespan assays performed at 25C? The standard temperature for the worm is 20C and here I think this is very relevant as the TOR pathway is responsive to suboptimal conditions. I wonder if the results are also true for lower temperatures.

****Minor comments:****

It would help to better delineate the rationale for the experiments in Fig. S1. Experiments here are aimed to find mediators of TOR effects distinct from Pol III. Such distinct mediators would be additive to Pol III (as is the case in the figure) and downstream of TOR.

***Strengths:** The study advances our knowledge regarding the timing of the Pol III targeting intervention for survival effects.

***Limitations:** The study relies only on genetic data and not all of it is conclusive.

This study will be interesting for the geroscience community with an eye on TOR inhibition and is relevant to worm biology.

I work with *C. elegans* as a genetic model and I am interested in protein homeostasis, metabolism, health, and longevity.

Response to Reviewers

Reviewer #1

*The manuscript reports effects on brood size, lifespan and healthspan upon manipulation of *C. elegans* genes encoding RagA, TOR and Pol III orthologs, as well as other well-characterized lifespan-affecting genes. The results point to complex relationships among TOR and Pol III that are not fully resolved, suggest a role for *rpc-1* Pol III that is additive with well-characterized lifespan pathways, indicate a late-life requirement for *rpc-1* Pol III to limit lifespan, and, contrary to a previous publication, suggest a muscle requirement for *rpc-1* Pol III for lifespan limitation.*

Major comments regarding key conclusions:

*The work demonstrates that brood size is reduced upon *rpc-1* Pol III RNAi feeding from the L4 stage. However, no further analysis is provided to show how later aspects of reproduction impair brood. Minimally, ruling out effects on spermatogenesis would be important since sperm number limits self-fertile brood size. It is also unclear from the methods whether the brood size results include embryonic lethality (post-reproduction). Internal hatching, if it occurred, could also affect interpretation of the results. A change in the reproductive period should be noted if it occurred.*

The reviewer is correct that it is important to address the role of Pol III more thoroughly in relation to reproduction. We have added several pieces of data to support our claims:

Point 1:

The question as to whether Pol III limits sperm function (or later developmental roles) is interesting and had not yet been addressed. To examine this we counted the spermatozoa in males treated with *rpc-1* RNAi at day 1 of adulthood as in Shakes et al., 2009, PLoS Genetics doi: [10.1371/journal.pgen.1000611](https://doi.org/10.1371/journal.pgen.1000611). However, *rpc-1* RNAi did not have any effect on sperm number. This data is now presented in **Figure S1A and B** and a section on Spermatocyte measurement has also been added to the methods.

Point 2:

In addition to the total brood size (live progeny) data shown initially, we have now quantified the total brood size (both eggs laid and hatched progeny) to address whether Pol III also impacts embryonic viability. We find that *rpc-1* RNAi reduces both the number of eggs laid and this equates to the number of eggs that hatch into live progeny. There are no dead eggs remaining. This data is shown in **Figure S1C**, which combined with the sperm counts shown in **Figure S1B** indicate that Pol III's effects are specific to eggs. We have also now established that *rpc-1* RNAi has no effect on the reproductive period of *C. elegans*. This age specific fecundity data is shown in **Figure S2A**. The following has been added to the results section to explain the data from points 1 & 2.

*“The reduction in brood size is likely independent of spermatogenesis as *rpc-1* RNAi did not diminish the total number of sperm in self-fertilized hermaphrodites (Fig. S1A and S1B). Nor did *rpc-1* RNAi treatment lead to any change in embryonic viability, or age-specific fecundity and embryos hatched into larvae within a similar time frame as WT (Fig. S1C and S2A). This suggests that Pol III inhibition from late larval development is sufficient to reduce progeny production without affecting spermatogenesis or embryo maturation ex-utero.”*

Point 3:

At 25°C (the temperature that we did these experiments) very few animals suffered internal hatching and those that did were taken out of the analysis – therefore this is unlikely to skew the results, as the reviewer suggests. A detailed analysis of the temperature effects of Pol III RNAi and lifespan is already published in our original Pol III paper, Filer et al., 2017 Nature. The following has been added to the methods section.

“At 25°C (the temperature that we did these experiments) very few animals suffered internal hatching and those that did were taken out of the analysis.”

The authors claim that, similar to the relationship previously concluded from aging studies, *rpc-1* acts downstream of TORC1. However, this claim is not well supported. In an effort to circumvent early lethality caused by loss of *let-363* ("CeTOR"), they use a mutation in *raga-1* *RagA* and demonstrate a further reduction in brood with *rpc-1* RNAi. If *raga-1(ok386)* were a null this result would demonstrate a relationship that is at least partially parallel, not linear. By contrast, double RNAi with *let-363* was "non-additive", suggesting a more linear relationship. However, interpretation of these experiments requires (1) that the *raga-1* mutation is null and affects only TORC1 signaling, (2) evidence that the double RNAi worked well (e.g., qPCR; see Ahringer et al. 2006 review regarding issues with multi-RNAi), and (3) failure to consider alternative effects of loss of *let-363* (e.g., TORC2). Negative results with RNAi are particularly problematic in the absence of convincing evidence that the RNAi worked well. Moreover, results in Figure 1G are difficult to interpret since the initial values are low. Here and elsewhere the genetics descriptions are unconventional, hampering interpretation. For example, what is meant by a mutation being "incomplete"? That it acts as a hypomorph?

We understand the concerns of the reviewer and thank them for giving us an opportunity to clarify things:

Point 4:

The *raga-1* mutant strain that we use is *raga-1(ok386)*. This allele harbours a 1242 bp deletion at the *raga-1* locus that removes almost the entire coding region of the gene. This effectively makes it null and dramatically downregulates TORC1 signalling. Details on this allele can be found on Wormbase, https://wormbase.org/species/c_elegans/variation/WBVar00091681#02-456-10. For reference, this strain/allele has been used in several other studies, e.g. doi.org/10.7554/eLife.49158. To add clarity to this manuscript the following has been added to the results section.

"The raga-1(ok386) strain harbors a mutation predicted to remove the entire gene, and is considered null."

...and this to the methods.

"The raga-1(ok386) allele harbors a 1242 bp deletion at the raga-1 locus that removes almost the entire coding region of the gene."

Point 5:

We agree that double RNAi can be challenging. To reduce the problem associated by double RNAi, we included controls where each RNAi control was also diluted 50:50 with control RNAi – in each case this was compared to non-diluted RNAi and phenotypes observed. We find that diluting either *rpc-1* RNAi or *let-363/CeTOR* RNAi reduces the brood size of *C. elegans* in a manner comparable to non-diluted RNAi. This suggests that the combinatorial RNAi will therefore target each molecule effectively. Additionally, we find that diluted RNAi has no effect on the numbers of eggs laid vs hatched larvae, similarly to the non-diluted RNAi. All this data is now presented in **Figures S2A-C**, with the following added to the results section.

"This was not due to an artifact of performing double RNAi as diluting either let-363/CeTOR or rpc-1 RNAi by 50% with EV control achieved the same fecundity-related phenotypes as undiluted RNAi (Fig. S2A-C)."

Point 6:

Our apologies that the nomenclature was confusing. The CeTOR RNAi nomenclature was 'borrowed' from other papers describing this tool e.g. doi.org/10.7554/eLife.31268 and [doi: 10.1371/journal.pgen.1000972](https://doi.org/10.1371/journal.pgen.1000972). However, to make our work clearer, we have changed ceTOR to ***let-363/CeTOR* RNAi** and *raga-1* to ***raga-1/RagA*** in the manuscript – as suggested by another reviewer (see below).

Another claim is that rpc-1 Pol III limits adult lifespan downstream of TOR. These results are not convincing. The two treatments (raga-1 mutation as "embryonic" and L4 stage "CeTOR" let-363 RNAi as late) are not directly comparable for reasons noted above, and the double RNAi problem hampers interpretation.

Point 7:

Our lifespan data points out that the longevity increase upon Pol III knockdown is additive with *let-363/CeTOR*, suggesting a mechanism independent of TOR. Indeed, due to lack of ideal reagents, we were forced to try the double RNAi knockdown approach for *TOR/let-363* and *Pol III/ rpc-1*. To make the data interpretation easier, and rule out the possibility of confounding background RNAi to the maximum possible extent, we have included appropriate RNAi controls. Wherever double RNAi has been used, the effect on the phenotype by 50% dilution of target RNAi with empty-vector control, has also been shown independently and used for the statistical comparison with combinatorial RNAi. Our results have shown that diluting *let-363* RNAi and *rpc-1* RNAi both to 50%, is enough to impart lifespan increase when initiated from L4 stage. All this data and statistical analysis is included in **Table 1**. In addition to the change described in point 5, a sentence has been added to the legend of Figure 2 to explain this more clearly and direct the reader to the relevant data.

“(A-C) Diluted and non-diluted RNAi treatments extend lifespan equally (Table 1). One representative experiment is shown, refer to Table 1 for data on all replicates and statistical analysis.”

*The nomenclature might be easier to follow if the authors state the actual *C. elegans* genes manipulated (e.g., *let-363 TOR* versus *raga-1 RagA*) rather than using "CeTOR" as a catch-all since these genes are not identical in action.*

Point 8:

Thank you for this suggestion. We have changed ceTOR to ***let-363/CeTOR* RNAi** and *raga-1* to ***raga-1/ RagA*** in the manuscript – as suggested by the reviewer (see point 6).

*Based on genetic interactions (*rsks-1, ife-2, ppp-1, daf-2* and germline loss) they show that *rpc-1* RNAi further extends the long lifespan conferred by each of the mutant alleles tested, as well as germline loss induced by two different mutant conditions. These results, though negative, are important. The statement that *rpc-1* does not affect global protein synthesis is somewhat overstated without additional experimental support.*

Point 9:

We thank the reviewer for supporting our inclusion of ‘negative data’. We agree that our statement relating to protein synthesis is overstated given the data presented. We will soften this to:

*“Taken together, these data show that *Pol III* knockdown does not seem to affect the lifespan incurred by reducing global protein synthesis, although this does not rule out the possibility that *Pol III* affect protein synthesis by other mechanisms not interact with global translation mechanisms.”*

*Extending and challenging their own previous work showing an intestinal focus of activity for *rpc-1* in limiting longevity (Filer et al., 2017), and noting that *RPC-1::GFP* detection can be knocked down by RNAi in several tissues, they use a tissue restricted *rde-1* expression approach (or *sid-1* for neurons) to test the contribution of intestine, hypodermis, neurons, muscle and germline. This new analysis points to a role for the muscle. This result is intriguing and warrants further experiments. To shore up tissue-specific claims the authors could consider (1) additional drivers for intestine and muscle *rde-1* in the RNAi experiments, or, ideally, a different approach such as tissue-specific protein degradation (again with multiple drivers), (2) a sufficiency experiment for muscle (wild-type muscle expression in the mutant to demonstrate reversal of the phenotype, or rescue of RNAi defects with an RNAi-insensitive reagent expressed in muscle).*

Point 10:

Thank you for appreciating the work we have done here and suggesting further experiments. To take your points one at a time: (1) We have already used the most robust tissue-specific alleles generated and reported in the *C. elegans* literature so far. It would be a significant amount of work to generate new *rde-1* driven tissue specific alleles to double check the Pol III levels/ *rpc-1* knockdown response in certain tissues, and we feel this is beyond the scope of this project. Suggestion (2) is interesting and would require us to generate a muscle specific *rpc-1* strain. However, there are issues with this approach. Firstly, it would require that we have a *rpc-1* mutant to rescue – which we don't as it is embryonically lethal and secondly it would not be possible to do this experiment using RNAi as the RNAi would then knock down the muscle construct.

*The possible explanation for the differences in *rde-1* results from the previous work should not be buried in the legends of Figure 3 and Figure S3. Perhaps this leaky background hypothesis should be directly tested (e.g., using the *RPC-1::GFP**

to examine whether residual expression exists in ne219 but not in ne300)? In any case, legend to Figure S3 needs editing: The ne219 background is not itself "intestine-specific", as implied, and the last sentence of Figure S3 legend should be "Thus, the rde-1(ne219)...", right?

Point 11:

The differences between the different tissue-specific strains is interesting. On reflection we agree with the reviewer that it should be included in the main text. We will describe the differences between the two rde-1 alleles ne219 and ne300 and our conflicting results in the discussion.

"Previously, we showed that rpc-1 RNAi in the intestine-specific RNAi strain VP303 significantly extends lifespan as reported in Filer et al., 2017. (Fig. S7). This observation is in contrast to our published data wherein rpc-1 RNAi extends the lifespan of a different intestine-specific RNAi strain rde-1(ne219). However, this difference is likely attributed to the allele-specific differences between rde-1(ne219) and rde-1(ne300); ne219 retains significant RNAi processing capacity and is not completely null whereas ne300 deletion allele is completely resistant to RNAi⁴². Thus, the rde-1(ne300) strain we initially tested is likely to be 'leaky' and RNAi knockdown might not be limited to the gut of the worms."

Finally, they show that late-adult rpc-1 RNAi extends lifespan over control RNAi and that, by several movement assays, healthspan is improved upon L4 rpc-1 RNAi, even when RNAi is active in muscle (based on WM118). The most significant new results are that rpc-1(RNAi) affects brood size, can extend lifespan (though modestly) after day 5 of adulthood, and that muscle may be involved rather than intestine.

Point 12 :

We agree with this summary.

Additional

comments:

Text throughout should clarify TOR vs presumed TORC1. Methods are insufficient. Important aspects of the lifespan methods and raw data are missing - e.g. exact numbers of worms censored. Exact information regarding statistical analysis is lacking (e.g., which tests, corrections for multiple testing). References should be given for all strains. For the rde-1 strains, it would be helpful to include, in addition to the transgene alleles, the actual promoters used to claim tissue specificity. Note, worms do not have "skeletal" muscle, as implied in the discussion. Figure 5 was not helpful for this reviewer. Figure legend to S3A is confusing: the intestinal signal appears stronger or at least equal, not weaker, in the rpc-1 RNAi background. Were these images collected using the exact same exposure settings?

To address this we have:

Point 13:

- Standardised genetic notation throughout the manuscript. CeTOR has been amended to let-363/CeTOR, and raga-1 to raga-1/RagA (see points 6 and 8) .

Point 14:

- Provided more detail on the specific promoters driving rde-1 for all transgenic alleles used. See legend **Figure 3A**.

"The intestine-specific, hypodermis-specific, germline-specific and muscle-specific RNAi rescue rde-1 in the rde-1(ne300) strain under mtl-2, col-62, sun-1 and myo-3 promoters, respectively."

Point 15:

- Added the CGC reference and grant information to the acknowledgements (as requested in their agreements) and wherever possible we have also supplied references for the stains we use.
- The Material and Methods section has also been revised to include the additional analyses. E.g.

"Spermatocyte measurement: Hermaphrodite C. elegans were raised and maintained as for brood size assays. At early L4 stage, 30-40 worms were moved to RNAi plates. After 24 hours (day 1 of adulthood) , worms were washed with 0.01% Tween PBS to remove excess bacteria, fixed in methanol for 5 minutes. Worms were washed once again with 0.1% Tween PBS. DAPI dye was added immediately and kept for 10 minutes. Worms were washed once again with 0.1% Tween PBS and mounted on 2.5% agarose pads for visualization. DIC and DAPI images were acquired using a Olympus IX 81 microscope. Individually distinct spermatocytes for each worm were counted."

Point 16:

- All statistical analyses performed are now described in the appropriate Legends.

Point 17:

- The lifespan methods has been revised to more accurately reflect our protocols (see below) and we have included censoring detail in each of the lifespan Tables for each individual experiment.

“Live, dead and censored worms were counted every 2–3 days in the worm populations by scoring movement with gentle prodding when necessary.”

Point 18:

- We have removed the reference to ‘skeletal muscle’ and replaced it with ‘body wall muscle’.

Point 19:

- The new data on the specific knockdowns and downstream targets achieved with *let-363 TOR* RNAi and *raga-1 RagA* mutation, as well as on the brood size/dead eggs effects, have been incorporated into Figure 5A for better clarity and readability.

Point 20:

- Brood size/embryo viability and age-specific fecundity data have been added and discussed in the manuscript. (See Supplementary Figs. S1 and S2)

Point 21:

- On reflection we agree that Figure S3A is confusing, mainly due to the gut autofluorescence in both the control and *rpc-1* RNAi conditions. We have amended **Figure S7** to make this clear and included additional images of each tissue to make it easier to see the tissue specific knockdown by RNAi.

Reviewer #1 Significance

See above. Study will be of interest to aging community.

Reviewer #2

The study by Malik and Silva et al describes results of the study investigating the role of RNA Polymerase III in regulating fecundity and lifespan in C. elegans. The authors show that knockdown of Pol III, similar to mTOR suppression, is detrimental for reproduction. Likewise, suppression of either Pol III or mTOR in adult animals extends lifespan via apparently the same pathway. In contrast, Pol III knockdown has an additive effect on lifespan in combination with other established genetic lifespan-extending approaches suggesting that they are working via different mechanisms. Furthermore, using the tissue-specific knockdown of Pol III the authors found that suppression Pol III expression in the muscle, but not other major worm tissues, is sufficient for its lifespan extending effect. Finally, the lifespan extension is also observed when Pol III knockdown is initiated late in adulthood. The overall conclusion is that suppression of Pol III expression late in animal life, particularly in the muscle, is a potential strategy to extend life- and health-span. Overall, the study is well-designed, the tools and results are robust and analysed appropriately. The data presentation is excellent, and the manuscript is clearly written. Addressing the points below will help to improve the clarity further.

We thank the reviewer for their very positive response to our study and are pleased that they found the data convincing. We are extremely pleased that the reviewer agrees with the design and tools used in this study. We can address all of the review's comments – as discussed below.

Major:

Significant amount of GFP signal is still present in RNAi treated animals, what is the tissue that maintains particularly high levels of expression (Fig. 3A) and how does it affect the conclusions? What is the level of Pol III reduction in different tissues? Could different efficiency of knockdown explain the tissue-specific effect of Pol III downregulation on lifespan? It would be important to show (and, if possible, to quantify) the knockdown efficiency in different tissues using the available reporter

Point 22:

We thank the reviewer for giving us the opportunity to improve and clarify this data. The *rpc-1::3xflag::gfp* reporter was used to a) determine the expression pattern of RPC-1 (original Figure 3A) and b) determine the effect of *rpc-1* RNAi on this (original Figure S5). We noted that RPC-1::GFP is expressed a wide number of tissues and when the reporter strain is treated with *rpc-1* RNAi, it is decreased in all tissues. Unfortunately, *C. elegans* has high levels of autofluorescence in the

intestine, attributable to lysozymes. To establish the tissue-specific efficiency of Pol III knockdown and also address the confounding issue of the autofluorescence we now present a new set of images in **Figure S7**, showing the impact of *rpc-1* RNAi on each tissue where Pol III is observed. We find that *rpc-1* RNAi is sufficient to completely remove expression in all tissues examined. In the case of the intestine, imaging the animals using a long-pass Dapi filter (which separates GFP (green) and autofluorescence (yellow)) it can now be seen that the nuclear expression of Pol III is gone in the presence of *rpc-1* RNAi.

Minor:

Fig. S3B is not cited in the text and the legend for the figure is somewhat confusing, potentially containing errors, this needs to be clarified.

Point 23:

We thank the reviewers for pointing this out. The figure has now been revised as a result of the analysis described above and is cited in the main text.

*“Previously, we showed that *rpc-1* RNAi in the intestine-specific RNAi strain VP303 significantly extends lifespan as reported in Filer et al., 2017. (Fig. S7). This observation is in contrast to our published data wherein *rpc-1* RNAi extends the lifespan of a different intestine-specific RNAi strain *rde-1(ne219)*. However, this difference is likely attributed to the allele-specific differences between *rde-1(ne219)* and *rde-1(ne300)*; *ne219* retains significant RNAi processing capacity and is not completely null whereas *ne300* deletion allele is completely resistant to RNAi (Watts et al.⁴²). Thus, the *rde-1(ne300)* strain we initially tested is likely to be ‘leaky’ and RNAi knockdown might not be limited to the gut of the worms.”*

Reviewer #2 Significance

*This is the first thorough study of Pol III knockdown as a lifespan extending strategy in *C. elegans*. In addition to the different laboratory model (previous study of Pol III in ageing primarily focused on *Drosophila*), this manuscript also offers several novel insights into consequences of Pol III perturbation at phenotypic, as well as mechanistic level in terms of interaction with other longevity pathways. The study will be of interest to those interested in processes underlying longevity and ageing. Considering that this topic is currently in fashion the publication will probably attract attention of not only specialist but also general public.*

We are extremely pleased that the reviewer shares our enthusiasm for this study and that they find the experimental evidence compelling.

Reviewer #3

Summary: *The paper by Yasir Malik et al investigates the genetic interrelationship between TOR signalling and Pol III expression regarding fecundity and longevity in *C. elegans*. Based on a previous study that defined a role of Pol III downstream of TOR in longevity across various species, this study looks particularly at the relative timing and tissue requirements for TOR and Pol III inhibition in longevity. Data indicate that Pol III acts downstream of TOR in regulating fecundity while there are additive effects regarding survival. The Pol III effect on longevity is based on its role in the muscle. Finally, health-span parameters mirror the survival data.*

Major comments: *This is a nice study the relies on genetic interaction to ask how TOR and Pol III interact. I find the observation that Pol III inhibition extends survival when initiated at day 5 of adulthood very exciting. In general, the study would benefit from additional data that back up the genetic observations.*

We thank the reviewer for appreciating the study and the novel insights it provides about the TOR-Pol III inter-relationship. We can address reviewer’s comments with the a few, limited experiments. Discussed below.

In Fig. 1, experiments are done to inhibit TOR to varying degrees in order to perform epistasis experiment. Of course these are difficult to interpret without the use of full KOs/loss of function. So while this is a good solution, it would be important to quantify the level to which TOR signalling is inhibited, optimally with biochemical experiments.

Point 24:

We fully appreciate the reviewer's point. Over the last few months we have been attempting a biochemical approach to address the impact of our manipulations on TORC1 activity, however this has been challenging and inconclusive. A similar concern was raised by reviewer 1 (point 5) and we have addressed this in two ways.

1) We agree that RNAi, and in particular double RNAi can be challenging. In the absence of biochemical data we have used a phenotypical analysis to assess the effectiveness of our genetic manipulations. To reduce the problem associated by double RNAi, we included controls where each RNAi control was also diluted 50:50 with control RNAi – in each case this was compared to non-diluted RNAi and phenotypes observed. We find that diluting either *rpc-1* RNAi or *let-363/CeTOR* RNAi reduces the brood size of *C. elegans* in a manner comparable to non-diluted RNAi. This suggests that the combinatorial RNAi will therefore target each molecule effectively. Additionally, we find that diluted RNAi has no effect on the numbers of eggs laid vs hatched larvae, similarly to the non-diluted RNAi. This data is now presented in **Figures S2A-C**, with the following added to the results section.

“This was not due to an artifact of performing double RNAi as diluting either let-363/CeTOR or rpc-1 RNAi by 50% with EV control achieved the same fecundity-related phenotypes as undiluted RNAi (Fig. S2A-C).”

2) We appreciate that this still does not provide a complete loss of function scenario so in addition we have softened our conclusions in relation to this aspect of the work.

2. General brood size is very low in the WT worms. Normally, one would expect 250-300 offspring per adult worm. It would be helpful if the authors could address this.

Point 25:

Indeed, as pointed out by the reviewer, the WT worms have a brood size of 250-300 eggs when kept at 20°C. but *C. elegans* exhibit different brood sizes dependent on temperature and these decline in size with increasing temperature. The experiments shown here were carried out at 25°C, where *C. elegans* produce less offspring. Our observation is in agreement with other studies of similar nature e.g. doi:10.1371/journal.pone.0112377 and doi.org/10.1371/journal.pone.0145925

3. Why were lifespan assays performed at 25C? The standard temperature for the worm is 20C and here I think this is very relevant as the TOR pathway is responsive to suboptimal conditions. I wonder if the results are also true for lower temperatures.

Point 26:

The reviewer raises an interesting point (and was also commented on by reviewer 1, point 3). This study follows from the previous study of Filer et al., Nature 2017 which demonstrated the role of Pol III in ageing. During this study we found and reported that there was a high proportion of intestinal bursting when lifespans were carried out at 20°C, which was ameliorated by carrying out the experiments at 25°C. This was explained and quantified in the original manuscript. To maintain consistency, we continued carrying out Pol III lifespans at this slightly higher temperature. Due to this limitation it is not possible to test the impact of TOR signalling on Pol III at lower temperatures.

Minor comments:

1. It would help to better delineate the rationale for the experiments in Fig. S1. Experiments here are aimed to find mediators of TOR effects distinct from Pol III. Such distinct mediators would be additive to Pol III (as is the case in the figure) and downstream of TOR.

Point 27:

Interpreting epistasis analysis is challenging. We were looking for interactors of Pol III using this targeted genetic approach and working on the premise that if two genes interacted then their effects would be non-additive. However, the reviewer is correct that if two genes are doing the same thing independently then their effects may be additive. Although our data does not suggest these mediators interact with Pol III in the same pathway to control lifespan (now shown in **Fig. S3**), it does not rule out the other possibility. We have clarified our rationale and softened our conclusion as follows...

“Reducing CeTOR signaling during development engages mediators distinct from Pol III to affect lifespan (Fig. 1C). These additional mediators may work downstream of CeTOR to drive organismal development. To identify these, we tested the genetic interaction of Pol III with candidate mediators implicated in protein synthesis or developmental processes.”

“Taken together, these data show that Pol III knockdown does not seem to affect the lifespan incurred by reducing global protein synthesis, although this does not rule out the possibility that Pol III affect protein synthesis by other mechanisms not interact with global translation mechanisms.”

Reviewer #3 Significance

Strengths: The study advances our knowledge regarding the timing of the Pol III targeting intervention for survival effects.

Limitations: The study relies only on genetic data and not all of it is conclusive.

This study will be interesting for the geroscience community with an eye on TOR inhibition and is relevant to worm biology. I work with C. elegans as a genetic model and I am interested in protein homeostasis, metabolism, health, and longevity.

April 15, 2024

RE: Life Science Alliance Manuscript #LSA-2024-02735

Dr. Jennifer MA Tullet
University of Kent
Unknown
CT2 7NZ
United Kingdom [GB]

Dear Dr. Tullet,

Thank you for submitting your revised manuscript entitled "Timing of TORC1 inhibition dictates Pol III involvement in longevity in *Caenorhabditis elegans*". We would be happy to publish your paper in Life Science Alliance pending final revisions necessary to meet our formatting guidelines.

- please be sure that the authorship listing and order is correct
- please upload all figure files as individual ones, including the supplementary figure files; all figure legends should only appear in the main manuscript file
- please add a Running Title and a Summary Blurb/Alternate Abstract to our system
- please add a Category for your manuscript in our system
- please add the Twitter handle of your host institute/organization as well as your own or/and one of the authors in our system
- please note that titles in the system on the manuscript file must match
- please consult our manuscript preparation guidelines <https://www.life-science-alliance.org/manuscript-prep> and make sure your manuscript sections are in the correct order and that they are labeled correctly
- please add Author Contributions to our system
- table S3 is missing
- please move your main, supplementary figure, and table legends to the main manuscript text after the references section
- we encourage you to revise the figure legend for Figure 1 such that the figure panels are introduced in alphabetical order

FIGURE CHECKS:

- please add scale bars to Figures 1B, 3A, 4E, S1A, S5 and S8

A. FINAL FILES:

B. MANUSCRIPT ORGANIZATION AND FORMATTING:

Sincerely,

Reviewer #2 (Comments to the Authors (Required)):

The manuscript investigates the role of Pol III in lifespan in worms. The authors show that suppression of Pol III expression late in animal life, particularly in the muscle, is a potential strategy to extend life- and health-span. My initial review was addressed by new experiments. The rebuttal refers to Figure S7 that does not exist in the updated manuscript (there are 4 supplementary figures). However I assume it is a typo and the authors mean Figure S3. Overall the manuscript has been improved and can be recommended for publication.

April 26, 2024

RE: Life Science Alliance Manuscript #LSA-2024-02735R

Dr. Jennifer MA Tullet
University of Kent
School of Biosciences
Giles Lane
Canterbury, Kent CT2 7NZ
United Kingdom

Dear Dr. Tullet,

Thank you for submitting your Research Article entitled "Timing of TORC1 inhibition dictates Pol III involvement in *Caenorhabditis elegans* longevity". It is a pleasure to let you know that your manuscript is now accepted for publication in Life Science Alliance. Congratulations on this interesting work.

DISTRIBUTION OF MATERIALS:

Again, congratulations on a very nice paper. I hope you found the review process to be constructive and are pleased with how the manuscript was handled editorially. We look forward to future exciting submissions from your lab.

Sincerely,
